# Cell type-specific driver lines targeting the *Drosophila* central complex and their use to investigate neuropeptide expression and sleep regulation

Tanya Wolff[1], Mark Eddison[1], Nan Chen[1], Aljoscha Nern[1], Preeti Sundaramurthi[2], Divya Sitaraman[2], Gerald M Rubin[1]*

[1]Janelia Research Campus, Howard Hughes Medical Institute, Ashburn, United States; [2]Department of Psychology, College of Science, California State University, Hayward, United States

## eLife Assessment

This is a **fundamental** body of work reporting anatomical, molecular, and functional mapping of the central complex in *Drosophila*. There were a few concerns of a minor nature, and all were addressed by the authors. The tools generated and the findings, which include characterization of neuromodulators used by different cells, will undoubtedly serve as a foundation for future studies of this brain region. The data are **compelling** and likely to have a major impact.

*For correspondence:
rubing@janelia.hhmi.org

Competing interest: The authors declare that no competing interests exist.

**Abstract** The central complex (CX) plays a key role in many higher-order functions of the insect brain including navigation and activity regulation. Genetic tools for manipulating individual cell types, and knowledge of what neurotransmitters and neuromodulators they express, will be required to gain mechanistic understanding of how these functions are implemented. We generated and characterized split-GAL4 driver lines that express in individual or small subsets of about half of CX cell types. We surveyed neuropeptide and neuropeptide receptor expression in the central brain using fluorescent in situ hybridization. About half of the neuropeptides we examined were expressed in only a few cells, while the rest were expressed in dozens to hundreds of cells. Neuropeptide receptors were expressed more broadly and at lower levels. Using our GAL4 drivers to mark individual cell types, we found that 51 of the 85 CX cell types we examined expressed at least one neuropeptide and 21 expressed multiple neuropeptides. Surprisingly, all co-expressed a small molecule neurotransmitter. Finally, we used our driver lines to identify CX cell types whose activation affects sleep, and identified other central brain cell types that link the circadian clock to the CX. The well-characterized genetic tools and information on neuropeptide and neurotransmitter expression we provide should enhance studies of the CX.

## Introduction

The central complex (CX) of the adult *Drosophila melanogaster* brain consists of approximately 2800 cells that have been divided into 257 cell types based on morphology and connectivity (*Scheffer et al., 2020*; *Hulse et al., 2021*; *Wolff et al., 2015*). These cell types are themselves organized into a set of highly structured neuropils (see *Figure 1*). The CX plays a key role in the flow of information between sensory inputs and motor outputs. It is particularly important in orientation and navigation, and much progress has been made in defining the cell types and circuits involved in these behaviors

**Figure 1.** Schematic diagram of CX neuropils. (**A**) The brain areas included in the hemibrain connectome are shown with the CX and key connected brain areas highlighted. (**B**) The neuropils comprising the CX are shown: FB, fan-shaped body; PB, protocerebral bridge; EB, ellipsoid body; NO, noduli; and AB, asymmetrical body. Redrawn from *Hulse et al., 2021*.

(reviewed in *Fisher, 2022*; *Green and Maimon, 2018*; *Pfeiffer, 2022*; *Turner-Evans and Jayaraman, 2016*). The CX also appears to participate in sleep and/or activity regulation (reviewed in *Dubowy and Sehgal, 2017*; *Shafer and Keene, 2021*). But these are unlikely to be the only functions performed by the CX.

The connectome has provided a detailed wiring diagram of the CX, information that will be critical for understanding how it performs its functions (*Hulse et al., 2021*). However, the functions of most cell types in the CX remain unknown. Gaining this knowledge will likely require measuring and manipulating the activity of individual cell types. Such experiments are greatly facilitated by the availability of cell type-specific genetic driver lines. Such lines also allow biochemical approaches for determining the neurotransmitters and neuropeptides used by individual cells. This is particularly relevant for the CX, which is one of the most peptidergic brain areas (*Kahsai and Winther, 2011*; *Nässel and Zandawala, 2019*). The roles of neuropeptide signaling in the CX are largely unexplored.

In this paper, we present the results of our efforts to develop and characterize genetic driver lines for CX cell types. We also provide a survey of neuropeptide gene expression in the adult central brain as well as determine neurotransmitter (NT) and neuropeptide (NP) gene expression in over 80 CX cell types. Many CX neurons express NPs, with some cell types expressing multiple NPs as well as a fast-acting neurotransmitter. Finally, we demonstrate the use of our collection of driver lines to screen for cell types that influence activity/sleep when activated. In doing so, we uncovered cell types not previously known to play a role in these processes as well as new pathways of communication between the circadian clock and the CX.

## Results and discussion
### Generation and analysis of split-GAL4 lines for CX cell types

We generated cell type-specific split-GAL4 lines for CX cell types using the same general approach that we previously applied to the mushroom body (*Aso et al., 2014*; *Meissner et al., 2023*; *Rubin and Aso, 2024*) and the visual system (*Tuthill et al., 2013*; *Wu et al., 2016*; *Nern et al., 2024*); see methods for details. In a previous report, we described the generation of split-GAL4 lines for cell types innervating the protocerebral bridge (PB), noduli (NO), and asymmetrical body (AB) (*Wolff and Rubin, 2018*). Here, we extend this work to the rest of the CX and include some improved lines for the PB, NO, and AB.

*Figure 2* and *Figure 2—figure supplements 1–4* show the expression of 52 new split-GAL4 lines with strong GAL4 expression that is largely limited to the cell type of interest. All lines were imaged

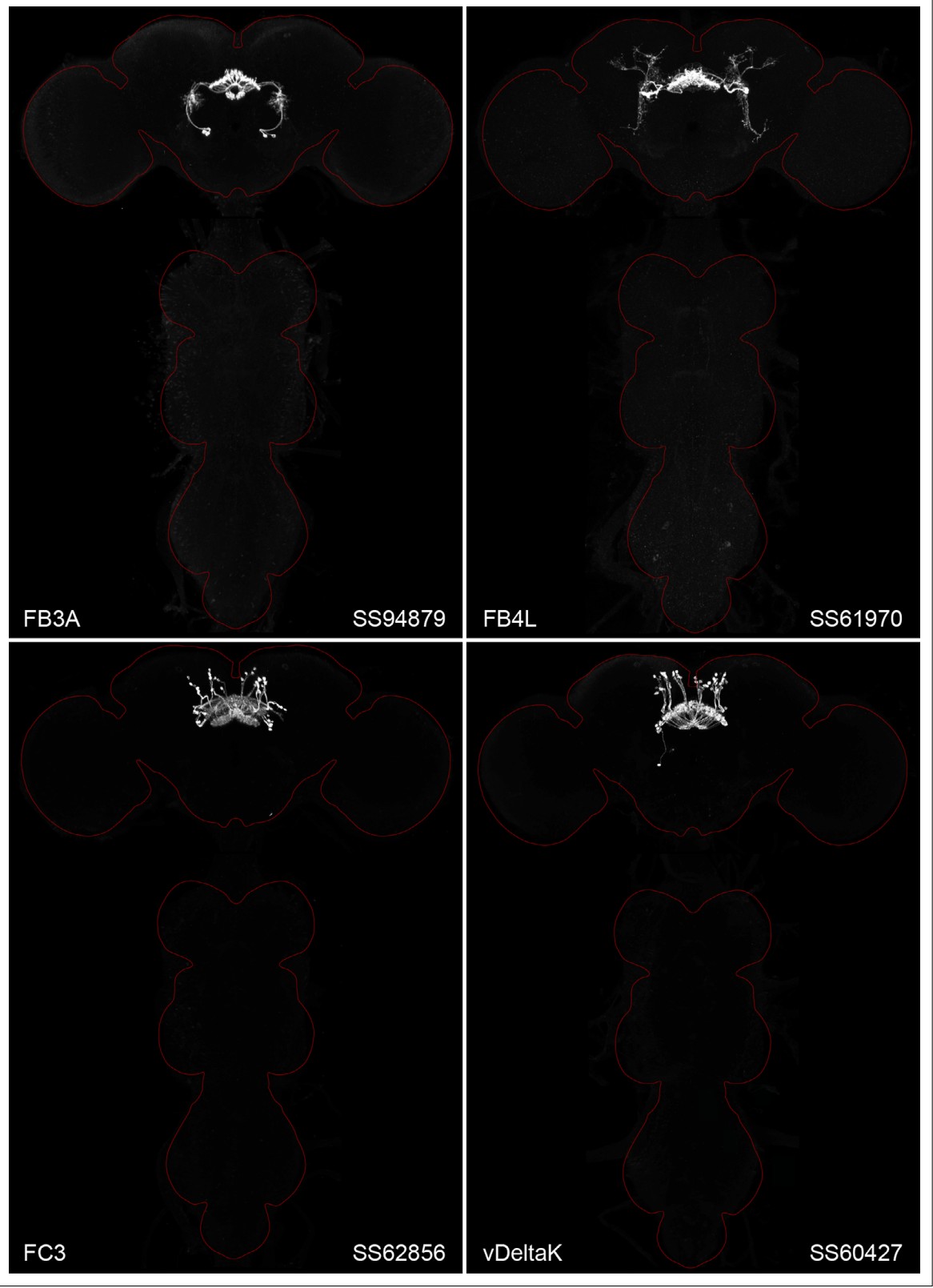

**Figure 2.** Maximum intensity projections (×20 confocal images) of the expression patterns driven by four stable split-GAL4 lines for the indicated cell types. The brain and VNC are outlined in red. Original confocal stacks that include a neuropil reference channel can be downloaded from https://www.janelia.org/split-gal4. Similar images for 48 additional lines generated as part of this study are shown in *Figure 2—figure supplements 1–4*. *Figure 2—source data 1* contains additional information on all the split-GAL4 lines we characterized.

*Figure 2 continued on next page*

*Figure 2 continued*

The online version of this article includes the following source data and figure supplement(s) for figure 2:

**Source data 1.** Table of split-GAL4 lines organized by CX structure and cell type.

**Figure supplement 1.** Maximum intensity projections (×20 confocal images) of the expression patterns driven by stable split-GAL4 lines for the indicated 12 cell types.

**Figure supplement 2.** Maximum intensity projections (×20 confocal images) of the expression patterns driven by stable split-GAL4 lines for the indicated 12 cell types.

**Figure supplement 3.** Maximum intensity projections (×20 confocal images) of the expression patterns driven by stable split-GAL4 lines for the indicated 12 cell types.

**Figure supplement 4.** Maximum intensity projections (×20 confocal images) of the expression patterns driven by stable split-GAL4 lines for the indicated 12 cell types.

in the brain and ventral nerve cord of adult females, and some were also imaged in males; we did not image expression in the peripheral nervous system or in non-neuronal tissues. Together with the other lines generated in this study and our previous work, we generated high-quality lines for nearly one-third of CX cell types that were defined by analysis of the connectome (*Hulse et al., 2021*). We also generated lines of lesser quality for other cell types that in total bring overall coverage to more than three quarters of CX cell types. These additional lines often show some combination of expression in more than one CX cell type, unwanted expression in other brain areas, or weak or stochastic expression. *Figure 2—source data 1* lists the two best split-GAL4 lines we generated for each CX cell type with comments about their specificity as well as the enhancers used to construct them. Additional split-GAL4 lines used in the sleep and NP/NT studies are also included in this file. Images of all lines are shown at https://www.janelia.org/split-GAL4 and the original confocal stacks of key imaging data can be downloaded from that site. For a subset of lines, images revealing the morphology of individual cells using MCFO (*Nern et al., 2015*), and higher resolution images, are also available (see e.g., *Figure 3*, *Figure 3—figure supplement 1*). Additional split-GAL4 lines that may be useful for further studies are listed in *Supplementary file 1*.

## Neurotransmitter expression in CX cell types

To determine what neurotransmitters are expressed in the CX cell types, we carried out fluorescent in situ hybridization using EASI-FISH (*Close et al., 2025*) on brains that also expressed GFP driven from a cell type-specific split-GAL4 line. In this way, we could determine what neurotransmitters were expressed in over 100 different CX cell types based on which members of a panel of diagnostic synthetic enzymes and transporters they expressed: for acetylcholine, ChAT (choline *O*-acetyltransferase; acetylcholine synthesis) and, in most cases, VAChT (vesicular acetylcholine transporter); for GABA, GAD1 (glutamate decarboxylase; GABA synthesis); for glutamate, vGlut (vesicular glutamate transporter); for dopamine, ple (tyrosine 3-monooxygenase; dopamine synthesis); for serotonin, SerT (serotonin transporter); for octopamine, Tbh (tyramine β-hydroxylase; converts tyramine to octopamine); and for tyramine, Tdc2 (tyrosine decarboxylase 2; converts tyrosine to tyramine) accompanied by lack of Tbh.

*Figure 4* shows two examples of this approach. In panels A–D, we provide evidence that the neurons comprising the PFGs celltype use a less common neurotransmitter, tyramine. Panels E–H show an example of apparent co-transmission. Here, the FB tangential neuron FB4K expresses RNAs suggesting it can release both acetylcholine and glutamate. Cases of co-transmission using two fast-acting neurotransmitters have been described in many organisms (reviewed in *Svensson et al., 2018*) including *Drosophila*, but are rare and may be post-transcriptionally regulated (*Chen et al., 2023*). Our full results are summarized, together with our analysis of neuropeptide expression in the same cell types, in *Figures 5–9*.

Methods for using machine learning to predict neurotransmitter from EM images show great promise (*Eckstein et al., 2024*). However, they are unlikely to fully replace the need for experimental determination and validation for three reasons. First, rarely used transmitters such as tyramine are problematic due to limited training data. Second, accurate prediction of co-transmission is challenging for current computational approaches. Third, FISH remains the gold standard for inferring gene expression.

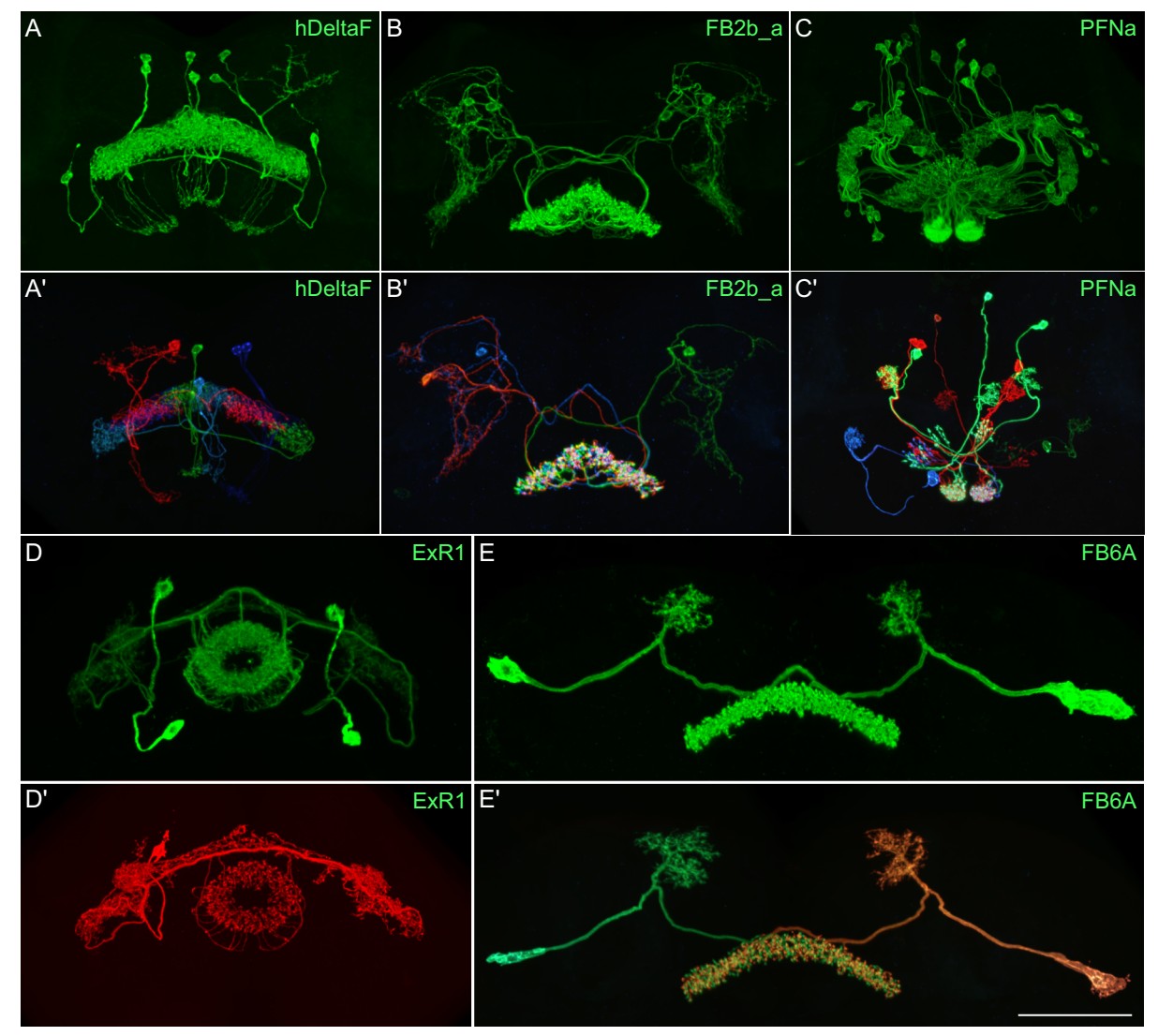

**Figure 3.** Visualization of the entire expression pattern of a split-GAL4 line for the indicated five cell types revealed using UAS-myrGFP (**A–E**). A subset of individual cells within those same cell types (A'–E') are revealed by stochastic labeling using the MCFO method (**Nern et al., 2015**). The scale bar in E' refers to all panels and = 50 µm. Images are maximum intensity projections (MIPs). Stable split lines used were as follows: A, SS54903; A', SS53683; B and B', SS49376; C and C', SS02255; D, SS56684; D', SS56803; E and E', SS57656. The original confocal image stacks from which these images were taken are available at https://www.janelia.org/split-GAL4.

The online version of this article includes the following figure supplement(s) for figure 3:

**Figure supplement 1.** Individual cell morphologies revealed by stochastic labelling.

## Survey of neuropeptide and neuropeptide receptor expression in the adult central brain

Neuropeptides provide a parallel mode of communication to wired connections and can act over larger distances using volume transmission (reviewed in **Nässel, 2009**; **Bargmann and Marder, 2013**). Neuropeptides are widely expressed in the CX (**Kahsai and Winther, 2011**; **Nässel, 2018**) and are likely to play important roles in its function. However, information on the expression of neuropeptides and their receptors is not provided by the connectome. To look for expression of neuropeptides in the CX, we took a curated list of 51 neuropeptide-encoding genes from FlyBase (FB2024_02, released April 23, 2024; **Öztürk-Çolak et al., 2024**) and eliminated 12 genes based on their not having been detected in RNA profiling studies of the adult brain. Trissin and Natalisin were added to the FlyBase list based on evidence summarized in **Nässel, 2018**. We only examined a small subset of neuropeptide

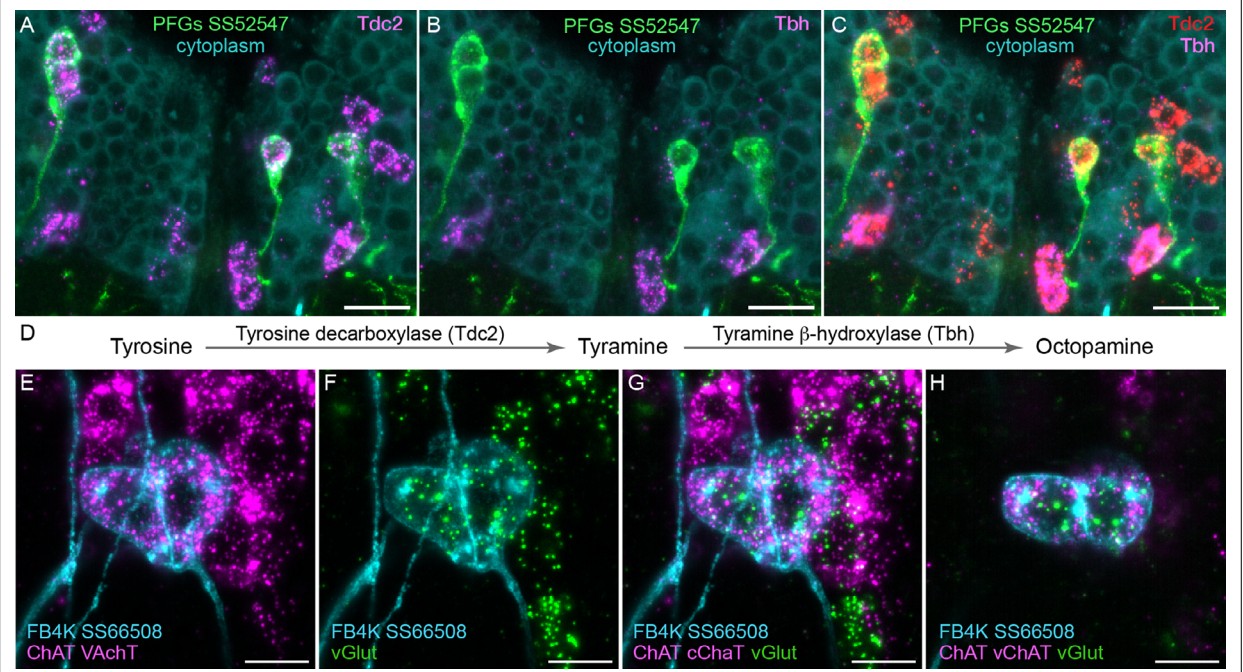

**Figure 4.** Using EASI-FISH to assess neurotransmitter expression. (**A–C**) Expression of the transcripts encoding key enzymes required to synthesize tyramine (Tdc2) and octopamine (Tbh) has been examined as indicated. PFG neurons were marked by the split-GAL4 line SS52547 together with UAS-myr-GFP and visualized by anti-GFP antibody staining. Brains were treated with DNAse1 and counterstained with high concentrations of DAPI to reveal total RNA in the cytoplasm. PFG cell bodies show expression of Tdc2 but not Tbh indicating that the PFG neurons use tyramine as a neurotransmitter. Maximum intensity projections (MIPs) of substacks are shown. (**D**) Biochemical pathway for synthesis of tyramine and octopamine from tyrosine. (**E–H**) Evidence for co-expression of the neurotransmitters acetylcholine and glutamate in FB4K. EASI-FISH was carried out using probes against choline acetyltransferase (ChAT) and vesicular acetylcholine transporter (VAChT) and vesicular glutamate transporter (vGlut) as indicated. The fan-shaped body tangential neuron FB4K has been visualized using the split-GAL4 line SS66508. Panels G and H show two different substacks at different Z-depths through the same neurons. Scale bar in each panel = 10 μm.

receptors, selecting those whose cognate neuropeptides we thought might play a role in the CX. We used EASI-FISH to determine the expression patterns of these genes in the adult central brain of females (*Figures 5–7*). The list of 41 neuropeptides and 18 neuropeptide receptors we explored is presented in *Figure 5—figure supplement 1*.

The neuropeptide expression patterns we observed fell into two broad categories. Some neuropeptides, like those whose expression patterns are shown in *Figure 5*, appeared to be highly expressed in only a few relatively larger cells. Such large neurosecretory cells often express the transcription factor DIMM (*Park et al., 2008*). Several of these neuropeptides—for example, SIFa (*Terhzaz et al., 2007*) and Dsk (*Wu et al., 2020*)—are expressed in broadly arborizing neurons that appear to deliver them to large areas of the brain and ventral nerve cord. In contrast, neuropeptides like those shown in *Figure 6* are expressed in dozens to hundreds of cells and appear poised to function by transmission to nearby cells in multiple distinct circuits. NPF and SIFa appear to act in both these modes. As we show below, most of the neuropeptides shown in *Figure 6* are expressed in the CX, each in distinct subsets of cell types.

Neuropeptide receptors (*Figure 7*) are more broadly, but not uniformly, expressed. In cases where more than one receptor has been identified for a given neuropeptide, such as Dh44 (*Figure 7G*; reviewed in *Lee et al., 2023*) and Tk (*Figure 7T*; see *Wohl et al., 2023*), the different receptors have distinct, but overlapping, expression pattens.

## Neuropeptide expression in CX cell types

We selected 17 neuropeptides whose transcripts were observed in cell bodies located in the same general brain areas as those of the intrinsic cells of the CX (see *Figure 5—figure supplement 1*). We used probes for these 17 genes to perform EASI-FISH on brains that also expressed GFP in a specific

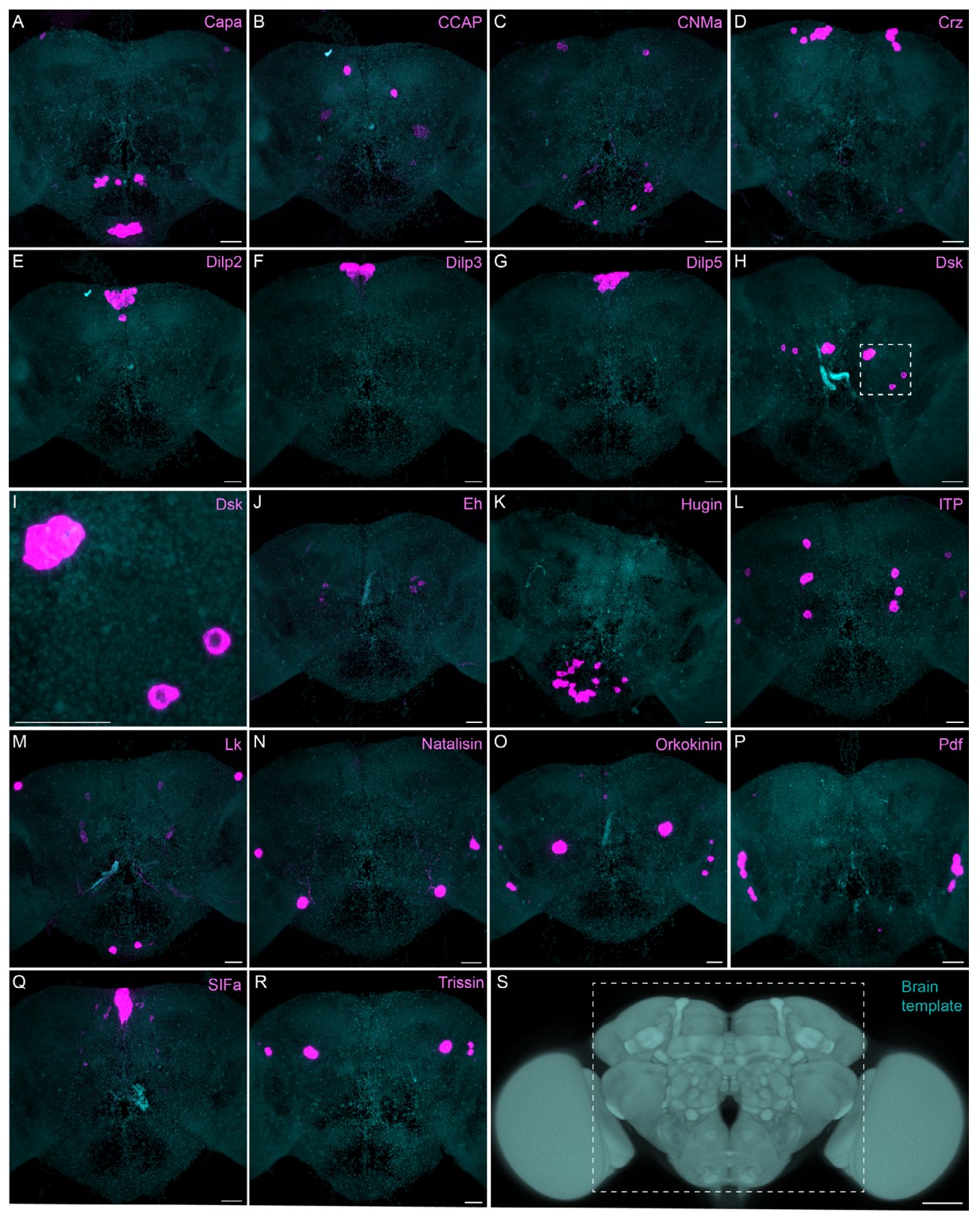

**Figure 5.** Sparsely expressed neuropeptide genes. (**A-R**) EASI-FISH was used to examine the expression of the indicated neuropeptide genes in the central brain (this brain area is shown as dashed box inS). Samples were counterstained with DAPI to visualize the outline of brain tissue. Images are MIPs. A higher magnification view of a region of the brain showing Dsk expression (indicated by dashed box in H) is shown in I. Scale bar in each panel = 50 µm; note that the images shown are from brains that were expanded by about a factor of two during the EASI-FISH procedure.

The online version of this article includes the following figure supplement(s) for figure 5:

*Figure 5 continued on next page*

*Figure 5 continued*

**Figure supplement 1.** Neuropeptides and neuropeptide receptor genes whose expression patterns were characterized by the EASI-FISH experiments shown in *Figures 5–8*.

cell type. In this way, we could score neuropeptide expression in individual cell types as we had done for neurotransmitters. *Figure 8* shows examples of this approach.

*Figure 9* presents a summary table of neurotransmitter and neuropeptide use by individual cell types based on our EASI-FISH results. *Figure 9—source data 1* contains a list of the individual stable split lines that were used for each cell type and how they were scored. We also characterized six CX cell types by RNA profiling (*Figure 9—figure supplements 1 and 2*) and additional RNA profiling of

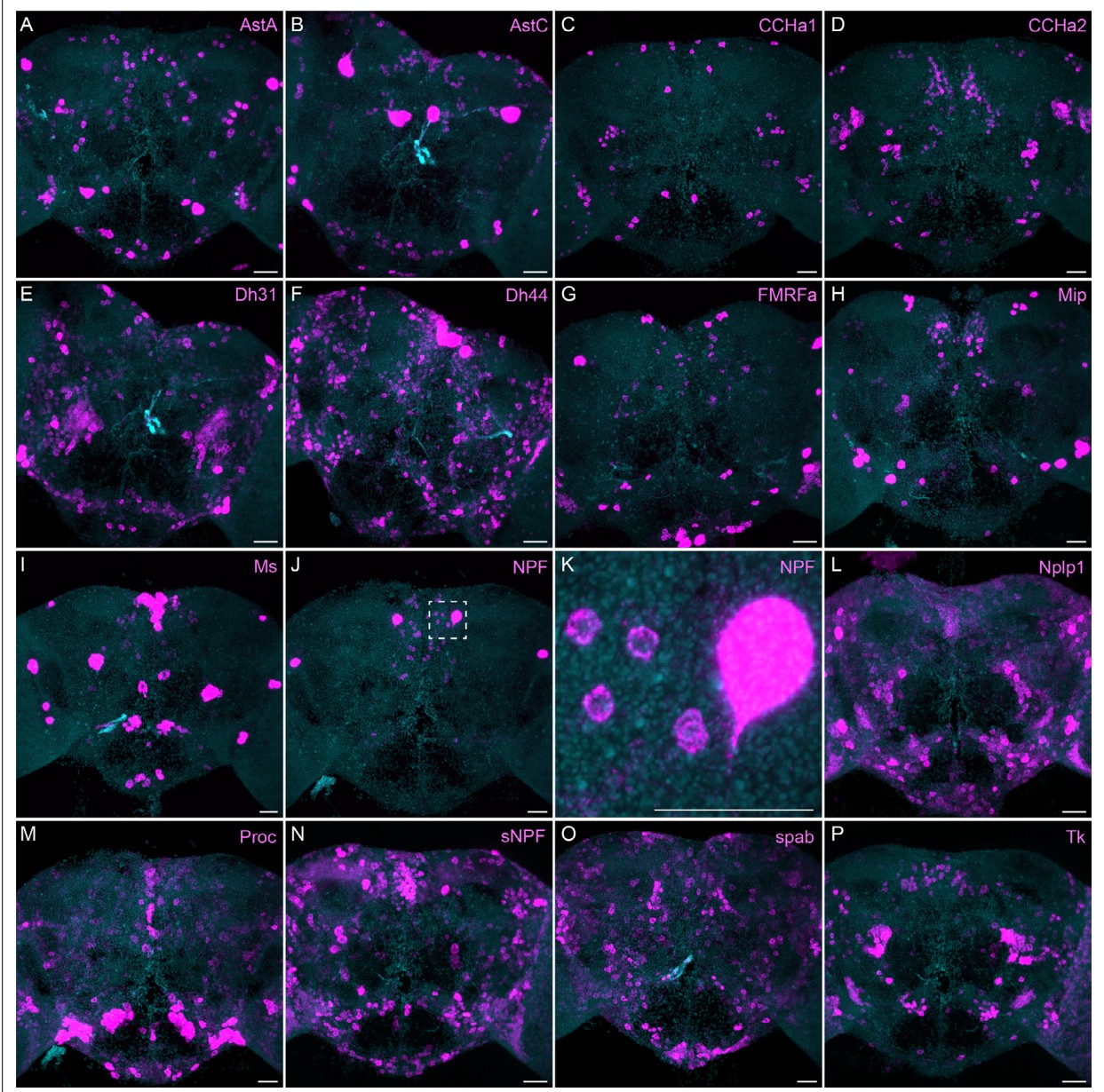

**Figure 6.** More broadly expressed neuropeptide genes. (**A-P**) EASI-FISH was used to examine the expression of the indicated neuropeptide genes in the central brain. Samples were counterstained with DAPI to visualize the outline of brain tissue. Images are MIPs. We included spab and Nplp1 in our screening although it is unclear whether these are indeed neuropeptides (M. Zandawala, pers. comm.). A higher magnification view of a region of the brain showing NPF expression (indicated by area enclosed by the dashed box in **J**) is shown in K. Scale bar in each panel = 50 μm; note that the images shown are from brains that were expanded by about a factor of two during the EASI-FISH procedure.

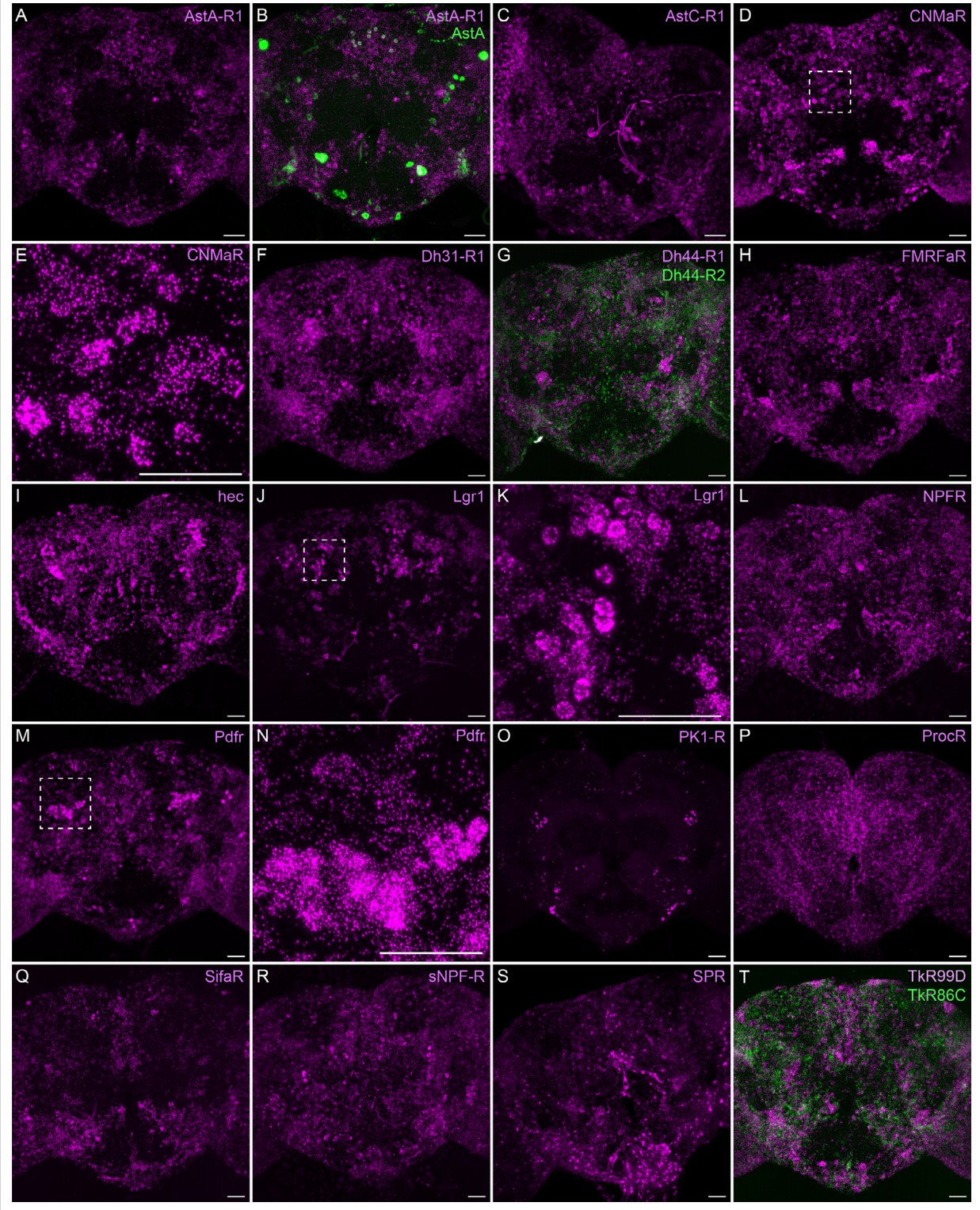

**Figure 7.** Neuropeptide receptor gene expression. (**A-T**) EASI-FISH was used to examine the expression of the indicated neuropeptide genes in the central brain. Higher magnification views of regions of the brains shown in D, J, and M (indicated by dashed boxes) are shown in E, K, and N. Images are MIPs. Scale bar in each panel = 50 μm; note that the images shown are from brains that were expanded by about a factor of two during the EASI-FISH procedure.

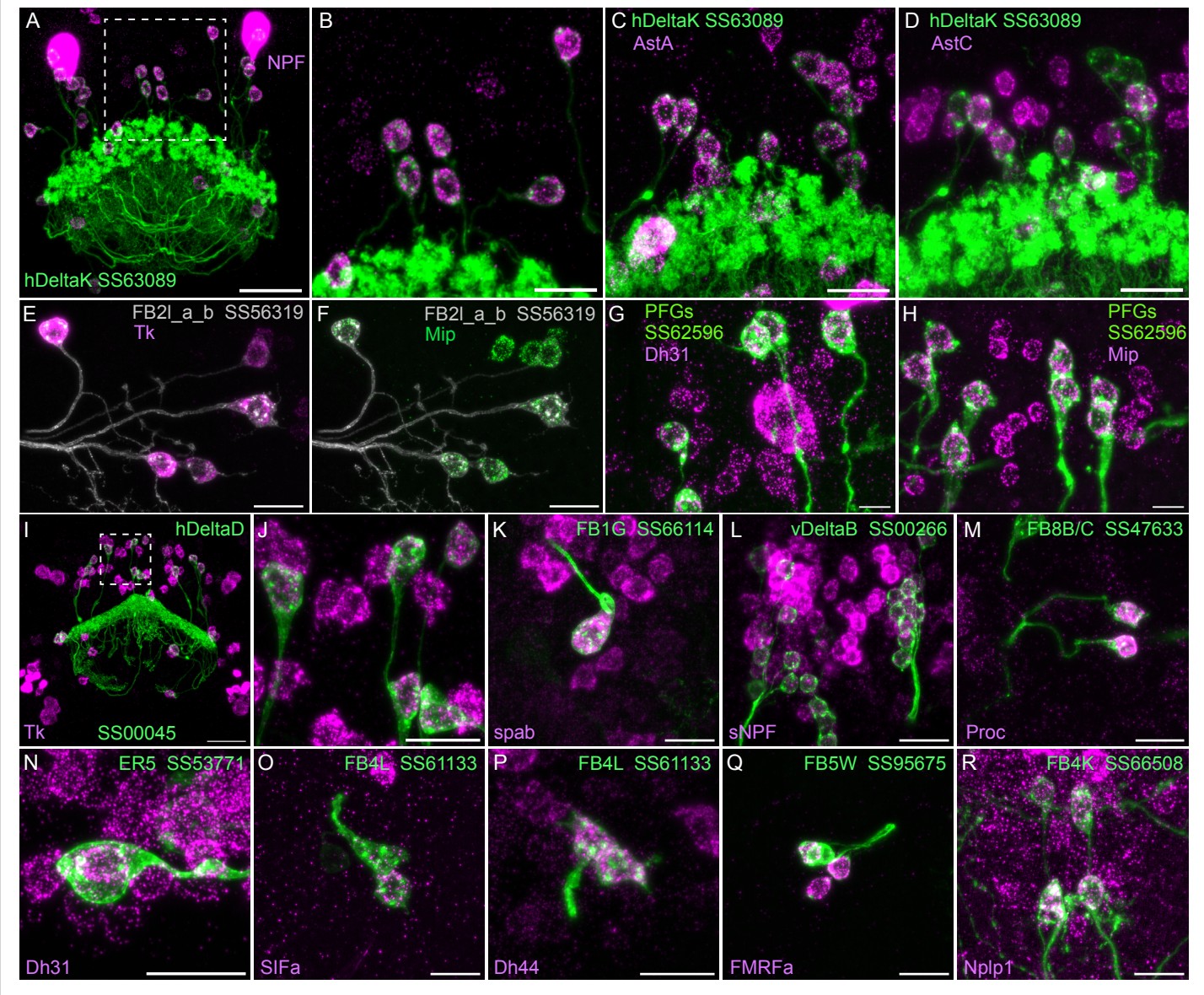

**Figure 8.** Neuropeptide gene expression in specific cell types. (**A-R**) EASI-FISH was used to examine the expression of the indicated neuropeptide genes in the brains also expressing myr-GFP driven by a cell type-specific split-GAL4 line. The line name and labeled cell type are indicated. GFP was visualized by anti-GFP antibody staining. *Hamid et al., 2024* demonstrated expression of Tk in ventral FB neurons likely to correspond to the cell type shown in panel E. Higher magnification view of regions of the brain in A and I (indicated by dashed box) are shown in B and J, respectively. Scale bar in panels A and I = 50 µm and for all other panels = 20 µm; note that the images shown are from brains that were expanded by about a factor of two during the EASI-FISH procedure.

CX cell types is provided in *Epiney et al., 2025*. These RNA profiling results are largely congruent with the EASI-FISH data and allow a comparison of transcript number and in situ signal strength.

A few general features emerge from these data. First, more than half of the cell types assayed express a neuropeptide. This frequency is perhaps not surprising given that the CX is considered one of the most peptidergic regions of the adult brain (*Kahsai and Winther, 2011*; *Nässel, 2018*); nevertheless, the fraction of cells expressing an NP appears to be several fold higher in the CX than observed in the adult brain as a whole, as judged by single-cell RNA profiling studies (*Davie et al., 2018*). Second, every cell type that expressed a neuropeptide also expressed a small molecule neurotransmitter (see *Nässel, 2018* for a discussion of other cases of co-expression). This co-transmitter was most often acetylcholine or glutamate, but we observed cases of GABA, dopamine, tyramine, octopamine, and

| cell type | Neurotransmitters | NT probes | Neuropeptides |
|---|---|---|---|
| PFNp+m | ChAT | 1,2,4 | Mip |
| PFNd | ChAT | 1 | Dh44 |
| PFNv | ChAT | 1,2,4 | Dh44 |
| PEN_a(PEN1) | ChAT | 1,2,4 | Dh44, FMRFa |
| PEN_b(PEN2) | ChAT | 1,2,5 | Dh44 |
| PEG | ChAT | 1,2,4 | none detected |
| EPG | ChAT | 1 | none detected |
| EPGt | ChAT | 1 | none detected |
| PFGs | Tdc2 | 1,2,3 | Mip, Dh31 |
| PFL1 | ChAT | 1,5 | — |
| PFL2 | Tdc2 | 1,2,4 | SIFa |
| LPsP | vGlut ple SerT | 1,5 | — |
| P6-8P9 | vGlut | 1 | — |
| Delta7 | vGlut SerT | 1,2 | Proc |
| P1-9 | vGlut | 1,5 | Proc |
| IbSpsP | ChAT | 1 | — |
| LNOa | vGlut | 1 | none detected |
| LCNOpm | vGlut | 1 | — |
| LCNOp/LCNp | vGlut | 1 | none detected |
| LNO1 | Gad1 | 1 | — |
| vDeltaA_a | ChAT | 1,5 | — |
| SA1_b | vGlut | 1 | — |
| hDeltaB | ChAT | 1,5 | — |
| hDeltaC | ChAT SerT | 1,2,3,5 | none detected |
| hDeltaD | ChAT | 1,2,3,5 | Dh44, Tk |
| hDeltaE | ChAT | 1,2,3 | Dh44, Tk, SIFA |
| hDeltaF | ChAT | 1,2,3 | none detected |
| hDeltaH | ChAT | 1 | Tk |
| hDeltaI | ChAT | 1,2,5 | AstA |
| hDeltaK | ChAT | 1,2,5 | AstA, NPF, AstC |
| hDeltaM | ChAT | 1 | — |
| vDeltaA_b | ChAT | 1,5 | — |
| vDeltaB | ChAT | 1,5 | sNPF |
| vDeltaC | ChAT | 1,2,3 | sNPF, Dh31 |
| vDeltaD | ChAT | 1,2,3,5 | Dh31, sNPF |
| vDeltaE | ChAT | 1,2,4 | Dh31, sNPF |
| vDeltaH | ChAT | 1 | — |
| vDeltaK | ChAT | 1,5 | — |
| vDeltaM | ChAT | 1,3,5 | Mip, Dh44 |

| cell type | Neurotransmitters | NT probes | Neuropeptides |
|---|---|---|---|
| FR1 | ChAT | 1,2,4 | sNPF |
| FC1 | ChAT | 1,2,4 | — |
| FC3 | ChAT SerT | 1,2,4 | — |
| FB1A | vGlut | 1,5 | none detected |
| FB1B | vGlut | 1,5 | none detected |
| FB1C | ple | 1,2,5 | Dh44 |
| FB1D | vGLuT ChAT | 1,2,4,5 | Mip |
| FB1F | vGlut | 1,5 | none detected |
| FB1G | ChAT | 1,5 | none detected |
| FB2A | ple | 1,2,3 | — |
| FB2B_a | vGlut Tdc2 SerT | 1,3,5 | Dh44, Proc |
| FB2B_b | vGlut Tdc2 SerT | 1,3,5 | Dh44, Proc |
| FB2C | vGlut | 1 | none detected |
| FB2E | vGlut | 1 | Mip, Tk |
| FB2F_a | vGlut | 1,5 | Dh31 |
| FB2G_b | vGlut | 1,5 | none detected |
| FB2H_a | vGlut | 1 | — |
| FB2H_b | vGlut | 1 | — |
| FB2I_a | vGlut | 1,5 | Mip, Tk |
| FB2I_b | vGlut | 1,5 | Mip, Tk |
| FB2J | vGlut | 1 | — |
| FB3A | vGlut | 1 | none detected |
| FB3C | Gad1 | 1 | Dh31 |
| FB3D | vGlut | 1,2,3 | Dh31 |
| FB4A | vGlut | 1 | Dh31 |
| FB4B | vGlut | 1 | Dh31 |
| FB4C | vGlut | 1 | — |
| FB4D | vGlut | 1 | none detected |
| FB4E | vGlut | 1 | Dh31 |
| FB4J | vGlut | 1 | none detected |
| FB4K | vGlut VAChT | 1 | Proc |
| FB4L | ple Tdc2 | 1,2,3 | AstC, Dh44, SIFA, Ms |
| FB4M | ple SerT | 1,2,3 | |
| FB4N | vGlut | 1 | none detected |
| FB4O | vGlut | 1 | Tk, Dh31 |
| FB4P_b | vGlut | 1 | Dh31, Tk |
| FB4Q_b | vGlut | 1 | — |
| FB4R | vGlut | 1 | Tk, Dh31 |
| FB4X | vGlut | 1 | none detected |
| FB4Y | SerT | 1,2,3 | none detected |
| FB4Z | vGlut | 1 | Dh44, Ms |

| cell type | Neurotransmitters | NT probes | Neuropeptides |
|---|---|---|---|
| FB5AB | ChAT | 1 | — |
| FB5B | vGlut | 1 | none detected |
| FB5G | vGlut | 1 | none detected |
| FB5J | none detected | | none detected |
| FB5K | vGlut | 1 | none detected |
| FB5L | vGlut | 1 | none detected |
| FB5N | vGlut | 1 | FMRFa, Proc |
| FB5Q | vGlut | 1 | none detected |
| FB5R | vGlut | 1 | Dh44, Ms |
| FB5T | vGlut | 1 | none detected |
| FB6A | vGlut ChAT | 1,5 | AstC |
| FB6B | vGlut | 1 | sNPF |
| FB6H | vGLuT ple SerT | 1,3,5 | — |
| FB6M | vGlut | 1 | Proc |
| FB6P | vGlut | 1 | — |
| FB6Q | vGlut | 1 | — |
| FB7A | vGlut | 1,2,3 | none detected |
| FB7B | vGlut ChAT ple SerT | 1,5 | Dh31 |
| FB7G | vGlut | 1 | — |
| FB7H | vGlut | 1 | Proc |
| FB7J | vGlut | 1 | Dh44 |
| FB7L | vGlut | 1 | none detected |
| FB8B | vGlut Tdc2 | 1,2,3,4 | Proc |
| FB8C | vGlut | 1 | Proc |
| FB8D | vGlut | 1 | none detected |
| FB8F_a | vGlut | 1 | none detected |
| FB8F_b | vGlut | 1 | none detected |
| FB8H | vGlut | 1 | none detected |
| FB9B | vGlut | 1,2,4 | — |
| EL | Tdc2 Tbh | 1,2,3 | SIFa |
| ExR1 | ChAT | 1,2,4 | none detected |
| ExR3 | SerT | 1,2,3,4 | none detected |
| ExR5 | vGlut | 1 | — |
| ExR6 | vGlut | 1 | — |
| ExR7 | ChAT | 1 | — |
| ER3d_a | Gad1 | 1 | — |
| ER4m | Gad1 ple | 1,2,4 | Dh31 |
| ER5 | Gad1 | 1,2,4 | Dh31 (PdfR Dh44R2) |
| ER6 | Gad1 | 1 | — |

**Expression level**

- **strong expression**
- weak expression
- very weak expression
- *expression only observed in a subset of GFP-expressing cells*

**Inputed neurotransmitters**

- ACh
- Glu
- GABA
- dopamine
- octopamine
- seratonin
- tyramine

**Figure 9.** Summary of neurotransmitter and neuropeptide expression as determined by EASI-FISH performed on adult brains in which selected cell types were marked by a split-GAL4 lines driving GFP expression. About half of all CX cell types were examined using probes for the following 15 NPs: AstA, AstC, CCAP, CCHa1, CCHa2, Dh31, Dh44, FMRFa, Mip, Ms, NPF, Proc, SIFA, sNPF, and Tk. All NPs except CCAP, CCHa1, and CCHa2 showed expression in at least one cell type. We excluded spab and Nplp1 from the genes in this figure, as it is unclear whether these are bonafide neuropeptides (M. Zandawala, pers. comm.); however, data on their expression in the listed cell types are given in *Figure 9—source data 1*. See *Figures 4 and 8* for examples of the experimental data supporting these conclusions. The specific split-GAL4 driver(s) used for each cell type and how they were scored are given in *Figure 9—source data 1*. Results are coded for signal strength by typeface as indicated. The color shading used for neurotransmitters indicates what we believe to be the most likely transmitter used by each cell type. 'None detected' indicates that an experiment was performed, whereas a '—' indicates no experimental data. To determine neurotransmitter expression various combinations of probe sets were used as indicated: 1, ChAT and VAChT choline *O*-acetyltransferase, and vesicular acetylcholine transporter; GAD1 (glutamate decarboxylase), vGlut (vesicular glutamate transporter); 2, ple (tyrosine 3-monooxygenase), SerT (serotonin transporter), Tbh (tyramine β-hydroxylase); 3, Tdc2 (tyrosine decarboxylase 2), Tbh; 4, Tdc2; and 5, SerT, ple, and Tdc2. In general, all lines were first probed with probe set 1 which reveals expression of genes involved in transmission by acetylcholine (ChAT), GABA (GAD), and glutamate (vGlut). Then a subset of lines was probed for genes involved in transmission by dopamine (ple), serotonin (SerT), octopamine (Tbh and Tdc2), and tyramine (Tdc2).

The online version of this article includes the following source data and figure supplement(s) for figure 9:

**Source data 1.** Table of split-GAL4 lines with EASI-FISH results for neurotrasmitter and neuropeptide expression.

**Figure supplement 1.** RNA-seq data for genes related to neurotransmitter synthesis, transport, and receptors in the indicated cell types.

**Figure supplement 2.** RNA-seq data for genes related to neuropeptides and their receptors in the indicated cell types.

serotonin. Third, co-transmission of two small molecule, fast-acting transmitters does occur but is rare. Conversely, co-transmission of a fast-acting transmitter and a modulatory transmitter such as serotonin is common: nine of ten cell types expressing the serotonin transporter also express another small transmitter, most often glutamate or acetylcholine. Octopamine is often, but not always, co-expressed with glutamate (see also *Sherer et al., 2020*).

## Screen for cell types whose activation modifies sleep

Sleep is a behavior widely studied in *Drosophila* (reviewed in *Dubowy and Sehgal, 2017*; *Shafer and Keene, 2021*). The phenotypic description of sleep and its relationship to activity, as well as the cell types that play a role in sleep regulation are under active study in many labs. The CX has been documented to be a significant brain region for sleep regulation. But many cell types in the CX have never been assayed for a role in sleep due to the lack of suitable genetic reagents. Therefore, we used our genetic drivers for CX cell types to screen for those whose activation by thermogenetics or optogenetics strongly influenced sleep or activity. As described in methods, we used three metrics: sleep duration; P(Doze), the probability that an active fly will stop moving; and P(Wake), the probability that a stationary fly will start moving. These assays were carried out over several years in parallel with our building the collection of lines, so many of the lines we assayed did not make it into our final collection of selected lines. Conversely, we did not assay all our best lines as many only became available after our behavior experiments were completed.

Our screen identified several cell types not previously associated with sleep and/or activity regulation. For example, hDeltaF was found to be strongly wake promoting (*Figure 10*). We also identified PEN_b (*Figure 10—figure supplement 1*), PFGs (*Figure 10—figure supplement 2*), EL (*Figure 10—figure supplement 3*), and hDeltaK (*Figure 10—figure supplement 4*) as likely to play a role. In most of these cases, we were able to assay multiple independent driver lines for the cell type. We also assayed several lines that each contained a mixture of dorsal FB cell types (*Figure 10—figure supplement 5*) but were otherwise free of contaminating brain or VNC expression. In addition to intrinsic components of the CX, we evaluated several cell types that, based on the connectome, we thought likely to convey information from the circadian clock to the CX. *Figure 11* (SMP368) and *Figure 11—figure supplement 1* (SMP531) present two such cases of strongly wake promoting cell types.

Results for lines not discussed in detail in the main paper are provided as Supplementary Files. *Supplementary file 2*, *Supplementary file 3* give results for 600 split-GAL4 lines assayed by thermogenetic activation with TRPA1 in both males and females, respectively. *Supplementary file 4* (males) and 5 (females) present results on over 200 lines, selected based on the results of thermogenetic activation, that were also assayed by optogenetic activation with CsChrimson. Images of the expression patterns of these lines and their genotypes can be found at https://flylight-raw.janelia.org/. The set of lines is anatomically biased with the dorsal FB overrepresented, and many lines contain multiple cell types and non-CX expression. Nevertheless, we believe these data might be useful as a starting point for further exploration.

The goal of our screen was to identify candidate cell types that warranted further study. While we identified several new potential sleep regulating cell types within CX, we did not perform the additional characterization needed to elucidate the roles of these cell types. For example, we did not examine the effects of inhibiting their function. Nor did we examine parameters such as arousal thresholds or recovery from sleep deprivation. Finally, by using only a 24-hr activation protocol we might have missed features only observable in shorter activation protocols. On the other hand, we assayed all lines in both males and females with identical genetic and environmental manipulations and many lines were evaluated by both optogenetic and thermogenetic activation. We observed that many lines showed phenotypes that differed between sexes, even though the expression patterns of the split-GAL4 lines did not obviously differ across sexes. Lines that showed strong effects generally did so with both activation modes and with both beam crossing-based activity measures and video-based locomotion tracking.

## Connections between the CX and the circadian clock

Not surprisingly, the connectome reveals that many of the intrinsic CX cell types with sleep phenotypes are connected by wired pathways. The strongest of these connections are diagrammed in *Figure 12*, with *Figure 12—figure supplement 1* also showing additional weaker connections. The connectome also suggested pathways from the circadian clock to the CX. Some of these have been previously noted. Links between clock output DN1 neurons to the ExR1 have been described in *Lamaze et al., 2018* and *Guo et al., 2018*, and *Liang et al., 2019* described a connection from the clock to ExR2 (PPM3) dopaminergic neurons. We found two SMP cell types, SMP368 and SMP531, that were very strongly wake promoting when activated suggesting they might be components of previously undescribed wired pathways from the clock to the CX.

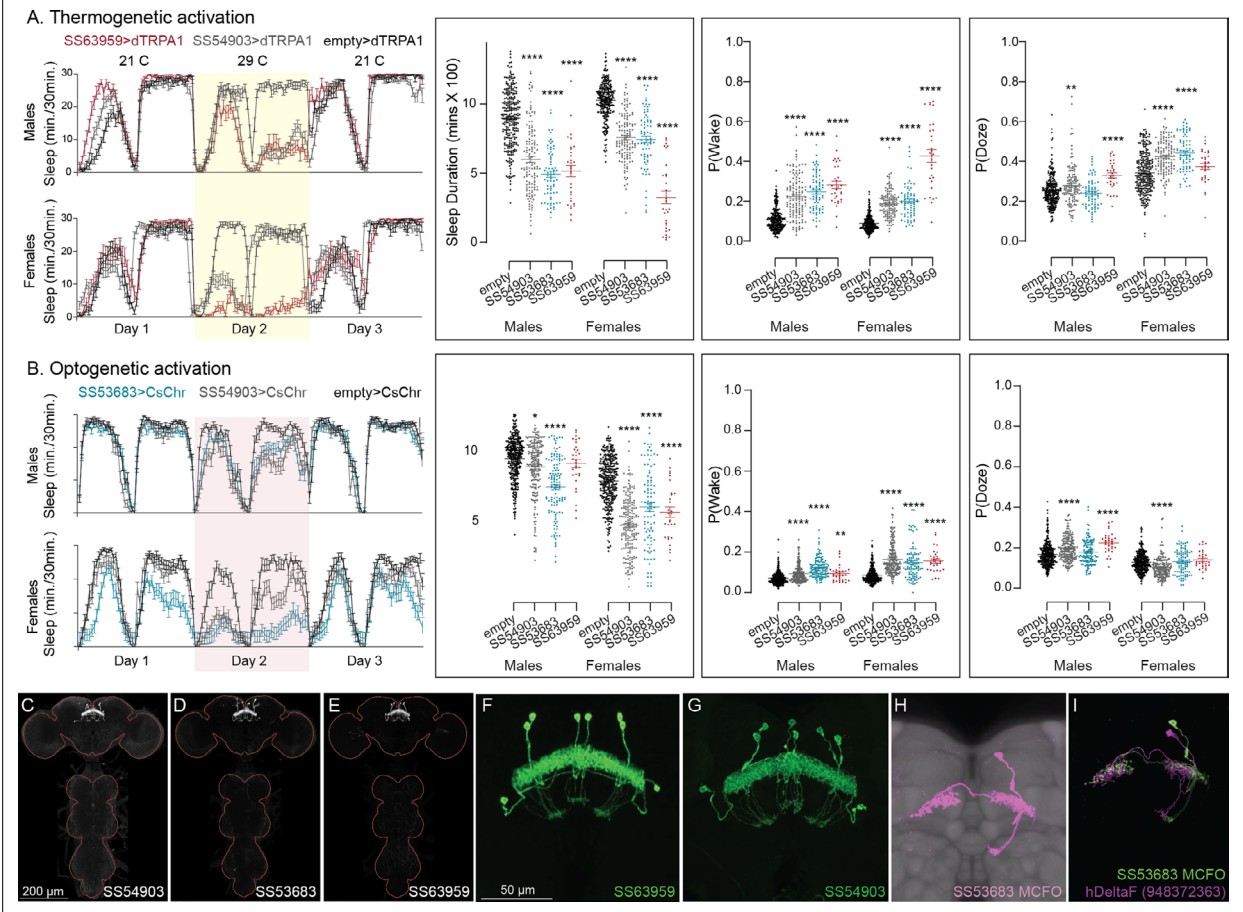

**Figure 10.** Activation of hDeltaF, comprised of eight intrinsic FB columnar neurons (see *Hulse et al., 2021*) decreases sleep. (**A**) Thermogenetic activation and DAM-based monitoring using the split-GAL4 lines SS54903, SS53683, and SS63959 produced decreased sleep duration and increased P(Wake) in all three lines as compared to the control (empty-GAL4). The sleep profile indicated a stronger suppression of sleep during nighttime in males and reduced sleep during both day and nighttime in females. (**B**) Optogenetic activation and video-based tracking showed that two split-GAL4 lines (SS54903 and SS53683) have decreased sleep duration, and all three tested lines have increased P(Wake) in male flies during optogenetic activation. All three lines showed decreased sleep and increased P(Wake) in female flies. As observed with thermogenetic activation, these phenotypes were more pronounced for nighttime sleep in males. In addition to sleep duration and P(Wake) we also measured activity by beam counts/waking minute in the DAM assays and pixel movements/waking minute in video tracking. We found that activation of hDeltaF does not increase these measures, showing that observed changes are not attributable to hyperactivity (see *Supplementary file 3*, *Supplementary file 4*, *Supplementary file 5*). (**C–E**) MIP images of GFP-driven expression in the brain and VNC of the three spilt-GAL4 lines. The brain and VNC are outlined in red. (**F**, **G**) Higher resolution images of the relevant brain area of two of the lines. (**H**) Morphology of a single neuron revealed by stochastic labeling. (**I**) Comparison of LM and EM cell morphologies. Original confocal stacks for panels **C–H** can be downloaded from https://www.janelia.org/split-gal4. The full genotypes of the driver lines are given there and in *Figure 2—source data 1*. Statistical comparisons were made by Kruskal–Wallis and Dunn's post hoc test. Asterisk indicates significance from 0: *p < 0.05; **p < 0.01; ****p < 0.0001.

The online version of this article includes the following figure supplement(s) for figure 10:

**Figure supplement 1.** Activation phenotype of PEN_b.

**Figure supplement 2.** Activation phenotype of PFGs.

**Figure supplement 3.** Activation phenotype of EL.

**Figure supplement 4.** Activation phenotype of hDeltaK.

**Figure supplement 5.** Activation phenotypes of combinations of dFB cell types.

In addition to these wired pathways, our work supports the possibility of signaling from the clock over considerable distances to the CX using neuropeptides. Our RNA profiling of ER5 cells (*Figure 9—figure supplement 1*), which are known to be regulators of sleep and sleep homeostasis (*Liu et al., 2016*), revealed expression of receptors for both PDF and Dh44. The presence of the PDF receptor in ER5 cells was suggested by prior work (*Im and Taghert, 2010*; *Parisky et al., 2008*; *Pírez et al.,*

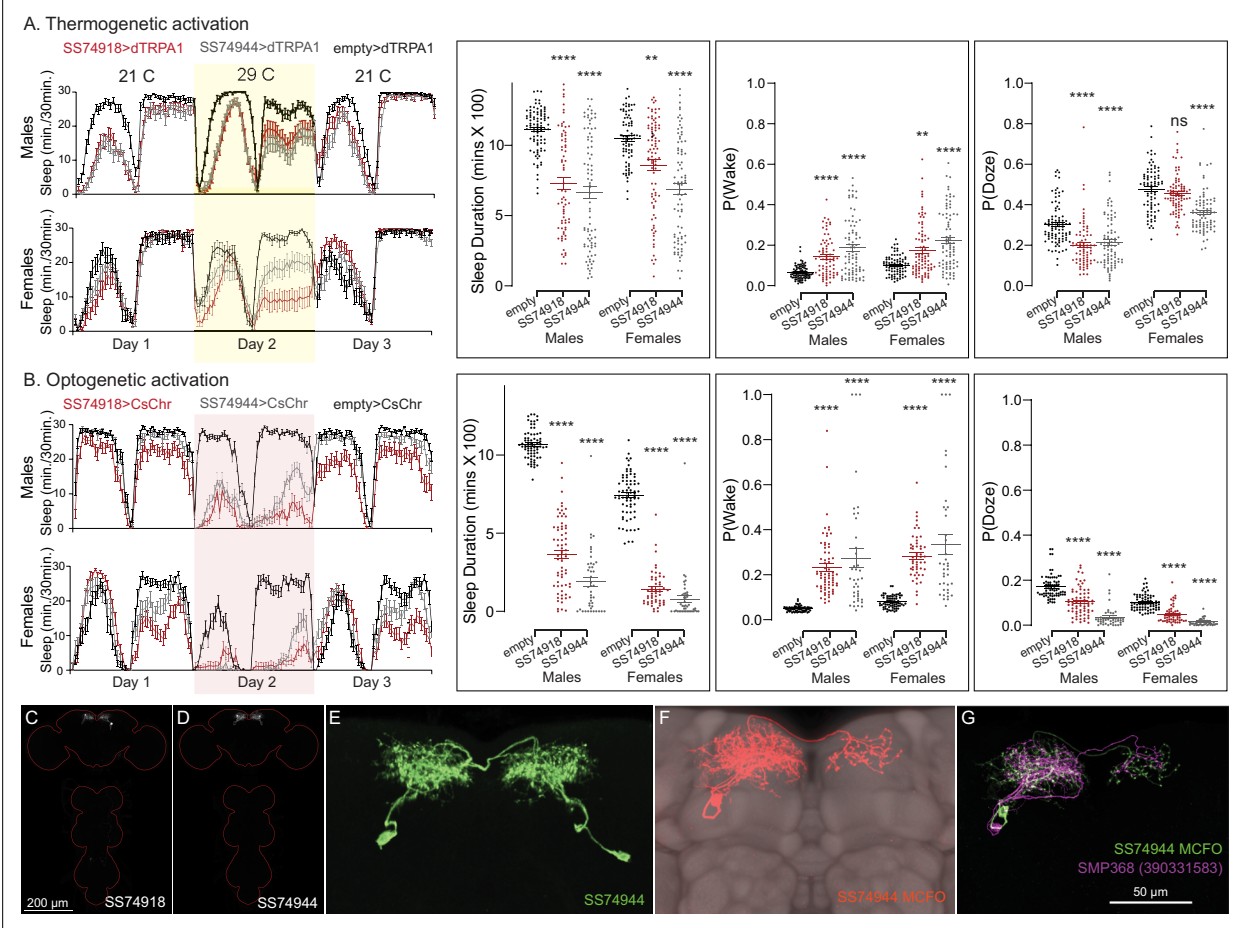

**Figure 11.** Activation of SMP368 decreases sleep. SMP368 connects central clock outputs to the CX (see *Figure 12*, *Figure 12—figure supplement 1*). Both split-GAL4 lines for SMP368 that we tested (SS74918 and SS74944) are strongly wake promoting in both thermogenetic (**A**) and optogenetic (**B**) activation and the phenotypes are consistent across sexes. Further, these lines show increase P(Wake) and decreased P(Doze) indicative of decreased sleep pressure and altered sleep depth. (**C, D**) MIP images of GFP-driven expression in the brain and VNC of the two spilt-GAL4 lines. The brain and VNC are outlined in red. (**E**) Higher resolution images of the relevant brain area of SS74944. (**F**) Morphology of a single neuron revealed by stochastic labeling shown with neuropil reference. (**G**) Comparison of LM and EM cell morphologies. Original confocal stacks for panels **C–F** can be downloaded from https://www.janelia.org/split-gal4. The full genotypes of the driver lines are given there and in *Figure 2—source data 1*. Statistical comparisons were made by Kruskal–Wallis and Dunn's post hoc test. Asterisk indicates significance from 0: **p < 0.01; ****p < 0.0001.

The online version of this article includes the following figure supplement(s) for figure 11:

**Figure supplement 1.** Activation phenotype of SMP531.

---

*2013*). We confirmed these observations and showed that ER5 cells make Dh31 (*Video 1*). Dh44 has been implicated as a clock output that regulates locomotor activity rhythms (*Barber et al., 2021*; *Cavanaugh et al., 2014*) and the DH44R1 receptor has been shown to function in sleep regulation in non-CX cells (*King et al., 2017*). However, the presence of the Dh44R2 receptor in the ellipsoid body (EB) was unexpected. Dh31 is expressed by many cells in the fly brain (see *Figure 6F*) including DN1s (*Kunst et al., 2014*) and has been shown to play a role in sleep regulation; the cellular targets of Dh31 released from ER5 are unknown, however previous work (*Goda et al., 2016*; *Mertens et al., 2005*; *Shafer et al., 2008*) has shown that Dh31 can activate the PDF receptor raising the possibility of autocrine signaling. *Andreani et al., 2022* also showed a functional link between the clock and ER5 cells, but the circuit mechanism was not elucidated.

## Concluding remarks

We provide a greatly enhanced set of genetic reagents for manipulating the intrinsic cell types of the CX that will be instrumental in fully elucidating the many functions of the CX. We illustrate their

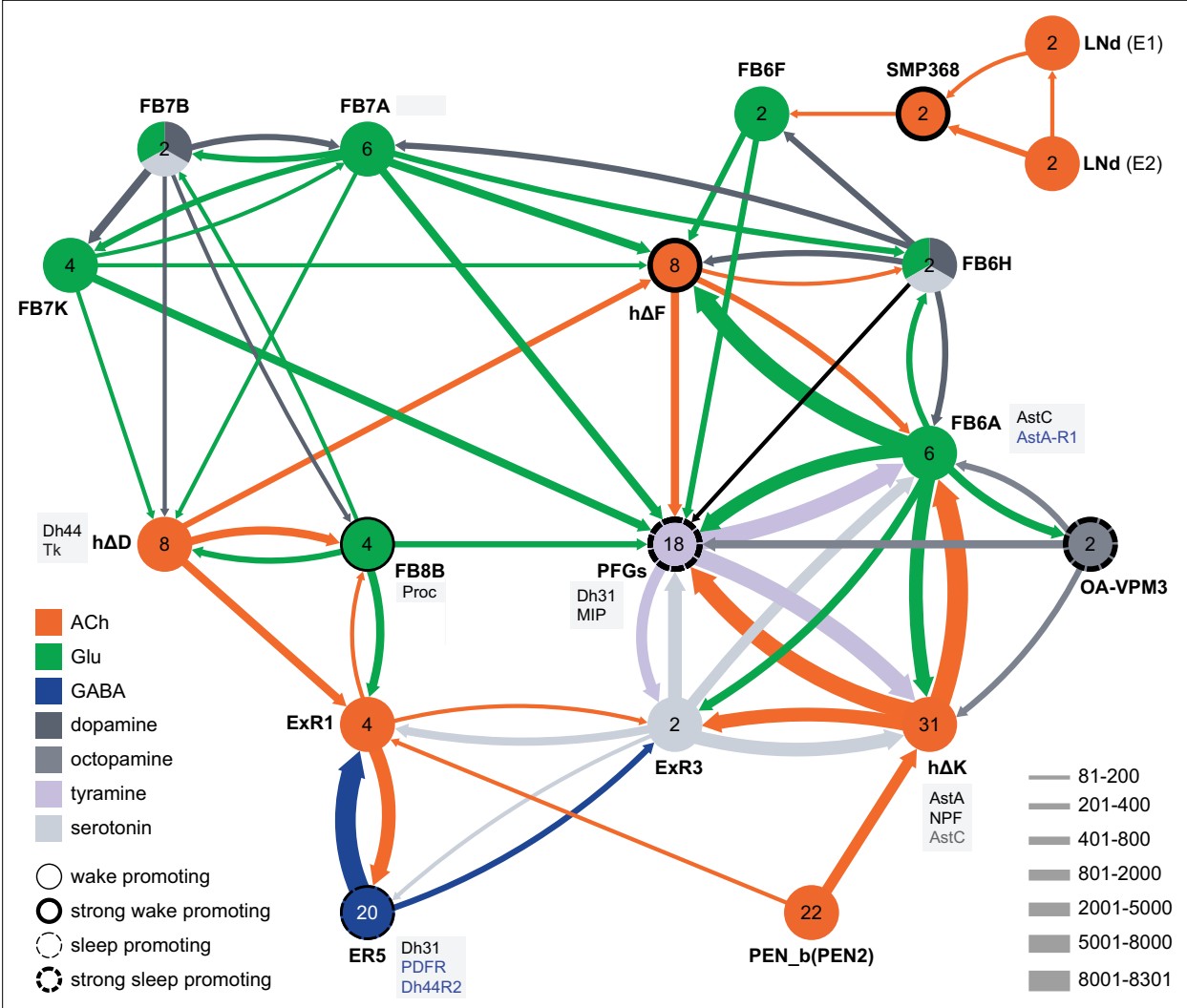

**Figure 12.** Circuit diagram of CX cells implicated in regulating sleep. Selected CX cell types, plus non-CX cell types SMP368, LNd, and OA-VPM3, are shown. The number within each circle denotes the number of cells in that cell type. The neurotransmitters used by each cell type are indicated by color coding, and the number of synapses between cell types is represented by arrow width. Cell types that have been shown to promote sleep or wake when activated are indicated. Experimental evidence for the wake promoting effects of OA-VPM3 is from Reyes, M and Sitaraman D (in preparation). In cases where we have experimentally determined expression of neuropeptides or neuropeptide receptor genes by either EASI-FISH or RNA profiling, this information is indicated in the boxes next to the relevant cell type. Cell type names are from the hemibrain release 1.2.1 except for the LNd neurons whose names have been modified based on morphology and connectivity; they have been grouped into two types: LNd (E1) corresponds to hemibrain body IDs 5813056917 + 5813021192 and LNd (E2) corresponds to hemibrain body IDs 511051477 (5th LNv) + 5813069648 (LNd6) (*Shafer et al., 2022*). Because the CX is a central body and the inputs from CX cells that have their soma in right or left hemisphere appear to be comingled on their downstream targets, the synaptic strengths shown represent the combined number of cells of each type, regardless of soma position. For example, the arrow thickness between FB6F and hDeltaF reflects the total number of synapses (368) from FB6F_R and FB6F_L to all hDeltaF cells; the individual synapse number between each of the two FB6F cells to each of the eight hDeltaF cells, which ranges from 7 to 39, can be found in neuPrint. The sole exception is the LNd cells where the synaptic strength represents only the output of LNds in the right hemisphere. See Figure 53 of *Hulse et al., 2021* for additional connected cells.

The online version of this article includes the following figure supplement(s) for figure 12:

**Figure supplement 1.** Circuit diagram showing some additional cell types potentially involved in regulating sleep.

use in discovering cell types involved in activity regulation, and uncovered new potential wired and peptidergic connections between the circadian clock and the CX. We surveyed neuropeptide and neuropeptide receptor gene expression in the adult central brain. Neuropeptides fell into two broad categories, those at are expressed in only a few cells and those that are expressed in dozens to hundreds of cells. We observed that neuropeptide receptor genes were much more broadly expressed

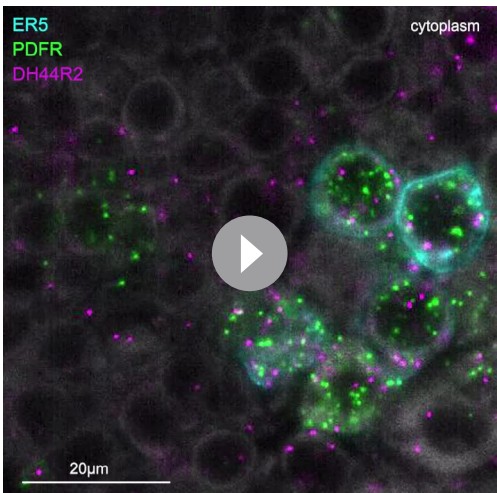

**Video 1.** Expression of genes encoding the neuropeptide receptors PDFR and Dh44R2 as well as the neuropeptide Dh31 in ER5 cells of the EB. The ER5 cells were marked by membrane-bound GFP expression and expression of PDFR, Dh44R2, and Dh31 were assayed by EASI-FISH.

https://elifesciences.org/articles/104764/figures#video1

than those of their cognate neuropeptides. Finally, we generated the largest available dataset of co-expression of neuropeptides and neurotransmitters in identified cell types. Unexpectedly, we found that all neuropeptide-expressing cell types also expressed a small neurotransmitter. Our data reveal the pervasive potential for peptidergic communication within the CX—more than half of the cell types we examined expressed a neuropeptide and one-third of those expressed multiple neuropeptides.

## Materials and methods
### Generation of split-GAL4 lines

Split-GAL4 lines were generated as previously described (*Dionne et al., 2018*). Databases of expression patterns generated in the adult brain by single genomic fragments cloned upstream of GAL4 (*Jenett et al., 2012*; *Tirian and Dickson, 2017*) were manually or computationally (*Meissner et al., 2023*) screened. Individual enhancers that showed expression in the desired cell type were then cloned into vectors that contained either the DNA-binding or activation domain of GAL4 (*Luan et al., 2006*; *Pfeiffer et al., 2010*). These constructs were combined in the same individual and screened for expression in the desired cell type by confocal imaging. Over 15,000 such screening crosses were performed to generate the new split-GAL4 lines reported here. Successful constructs were made into stable lines.

The lines listed in *Figure 2—source data 1* are currently being maintained at Janelia and the majority of these have also been deposited in the Bloomington *Drosophila* Stock Center.

### Characterization of split-GAL4 lines

Lines were characterized by confocal imaging of the entire expression pattern in the brain and VNC at ×20. Most lines were also imaged at higher magnification (×63) and/or subjected to stochastic labeling (MCFO; *Nern et al., 2015*) to reveal the morphology of individual cells. Split-GAL4 images are shown as MIPs after alignment to JRC2018 (*Bogovic et al., 2020*). Over 1800 confocal stacks derived from over 450 lines generated during this work are presented in, and can be downloaded from, an on-line database (janelia.org/split-GAL4). Images for the additional lines used in the sleep screen are available from flylight-raw.janelia.org.

Determining the correspondence between the cell types present in each split-GAL4 line and those described in the connectome (*Hulse et al., 2021*) was based solely on morphology. Even when assigning correspondence between cells in two different connectomes, where information on connectivity can also be employed, the process is not always straightforward (see *Schlegel et al., 2024*). Because of the similarity in morphology of many of the CX cell types it was often challenging to assign correspondence to the cell types defined by connectome analysis. For this reason, we rated our confidence in our assignments as Confident, Probable, or Candidate and include this information for each line at janelia.org/split-GAL4janelia.org/split-GAL4. To be considered confident, we judged our opinion had a >95% chance of being correct. Such assignments were generally only possible for cell types which had morphological features clearly distinct from those in other cell types. Most assignments were rated as Probable indicating 70–95% confidence. Lines whose cell type assignments are listed as Probable have been rigorously examined and the assignments are the most accurate that the available data allow. In the absence of single-cell data available in MCFO brains (the case for many lines) or additional data (e.g., physiological data on connectivity), cell types that are morphologically

very similar cannot be distinguished with complete confidence. Lines whose cell type assignments are listed as Candidate are even less certain (30–70% confidence).

## RNA in situ hybridization

Adult females (5–7 days post-eclosion) were expanded, probed, and imaged using the EASI-FISH method as described in *Eddison and Ihrke, 2022* and *Close et al., 2025*. The oligo probes and HCR hairpins were designed by, and purchased from, Molecular Instruments, Inc Imaging was performed on a Zeiss Z7 microscope equipped with a ×20 objective. Laser power and exposure time were optimized to maximize the signal-to-noise ratio. For neurotransmitters, the specific probes used for each cell type are indicated in *Figure 9*. For neuropeptides, each of the 17 selected NP probes shown in *Figure 5—figure supplement 1* was used on all cell types in *Figure 9* except those marked by '—' in the neuropeptide column.

## RNA profiling

The data shown in *Figure 9—figure supplement 1* were generated as described in *Aso et al., 2019*. See NCBI Gene Expression Omnibus (accession number GSE271123) for the raw data and additional details.

## Sleep measurement and analysis: thermogenetic activation screen

Split-GAL4 flies were crossed to 10× UAS-dTrpA1 (attP16) (*Hamada et al., 2008*) and maintained at 21–22°C in vials containing standard dextrose-based media (7.9 g agar, 27.5 g yeast, 52 g cornmeal, 110 g dextrose, 8.75 ml 20% Tegosept, and 2 ml propionic acid/l).

Virgin female progeny or male progeny (as specified in the figures), 3–7 days post-eclosion (*n* = 16–32/trial) were placed in 65 mm × 5 mm transparent plastic tubes with standard cornmeal dextrose agar media and placed in a *Drosophila* Activity Monitoring system (Trikinetics). Food composition was kept consistent between rearing and experimentation. Locomotor activity data were collected in 1 min bins for 5–7 days. Activity monitors were maintained with a 12-hr:12-hr light–dark cycle at 40–65% relative humidity. Total 24 hr sleep amounts (daytime plus nighttime sleep duration), Pwake, and Pdoze were extracted from the locomotor data as described in *Donelson et al., 2012*; *Wiggin et al., 2020* using MATLAB-based SCAMP program (*Sitaraman et al., 2024*; *Vecsey et al., 2024*).

Sleep duration was defined as 5 min or more of inactivity (*Hendricks et al., 2000*; *Shaw et al., 2000*). Representative sleep profiles were generated representing average (*n* = 24–32) sleep (min/30 min) for Day 1 (baseline), Day 2 (activation), and Day 3 (recovery/post activation). In addition to permissive temperature controls, split-pBDPGAL4U /dTrpA1 were used as genotypic controls for hit detection. pBDPGAL4U (attP40, attP2), has enhancerless GAL4-AD construct and GAL4-DBD constructs inserted on chromosomes II and III (*Dionne et al., 2018*), as is the case for split-GAL4 driver lines in behavioral assays. Each split-GAL4 line was tested at least twice in independent trials. For all screen hits, wake activity was calculated as the number of beam crossings/min when the fly was awake. Statistical comparisons between experimental and control genotypes were performed using Prism (GraphPad Inc) by Kruskal–Wallis one-way ANOVA followed by Dunn's post-test. Pairwise comparisons between the empty (split-pBDPGAL4U) control and experimental lines were made using Mann–Whitney *U*-test.

## Sleep measurement and analysis: optogenetic activation screen

Split-GAL4 flies were crossed to 20XUAS-CsChrimson-mVenus-trafficked (attP18) (BDSC:55134) and maintained at 21–25°C in vials containing standard cornmeal food supplemented with 0.2% retinal. Male and virgin female progeny were collected into separate vials containing standard cornmeal food with 0.4% added retinal and kept in a 25°C incubator on a 12:12 light:dark schedule for 3–5 days before loading. Typical sleep experiments lasted 6–7 days. Flies were loaded into 96-well plates or individual tubes using $CO_2$ anesthesia. Flies were allowed to recover from anesthesia and acclimatize to the experimental chambers for 16–18 hr prior to starting the experiment.

The behavioral setup for video recording system was adapted from *Guo et al., 2018*. Flies were briefly anesthetized and loaded into 96-well plates (Falcon 96-Well, Non-Treated, Fisher Scientific Inc) containing 150 ml per well of 5% sucrose, 1% agarose, and 0.4% retinal. The plates were covered with breathable sealing films. Small holes (one per well) were poked with fine forceps into the film to further ensure air exchange and prevent condensation. The entire setup was housed in an incubator to control

light/dark conditions and temperature. The 96-well plates with flies were placed in holders, constantly illuminated from below using an 850-nm LED board (Smart Vision Lights Inc) and imaged from above using a FLIR Flea 3 camera (Edmund Optics Inc). 635 nm red light (for optogentic activation) was provided using an additional backlight, low levels of white light (to provide a light–dark cycle) were supplied from above. Optogenetic activation was for a 24-hr period (starting in the morning at the same time as the white light was turned on for the day) and was delivered in pulses of 25-ms duration at 2 Hz frequency. Each experiment also included at least one full day without the red light preceding and following the activation day (matching the general design of the thermogenetic experiments).

Fly movement was tracked using single fly position tracker (GitHub - cgoina/pysolo-tools; *Goina, 2024*) and processed for sleep duration and other sleep parameters using SCAMP. In addition to the 5 min criteria, used to define total duration of sleep, P(Doze), the probability that an active fly will stop moving, and P(Wake), the probability that a stationary fly will start moving provide key additional measures of inactivity and activity and were included in our analyses (*Wiggin et al., 2020*). These sleep measures are presented in *Figures 10 and 11* (and supplements) and *Supplementary file 3*, *Supplementary file 4*, *Supplementary file 5*.

## Supplementary files for sleep phenotypes

The supplementary files present data on Day 1 (baseline) and Day 2 (activation). Given the large effects of environmental conditions on activity, we calculated p values between experimental and control group within the same environmental conditions. However, we also present mean differences between the days as a complementary way to identify lines that modified sleep when activated.

## Acknowledgements

We thank Y Aso (SS32244), J Goldammer (SS51128), and M Ito (SS49931, SS48762) for providing GAL4 lines. Robert Ray performed the RNA profiling experiments summarized in *Figure 9—figure supplements 1 and 2* and the meta-analysis of RNA profiling data to identify which neuropeptide genes were expressed in the adult brain. Marisa Dreher (Dreher Design Studios) performed connectomic analyses and figure generation. We thank Janelia's Project Technical Resources led by Gudrun Ihrke for assistance: ME, Kari Close and Yisheng He performed EASI-FISH experiments, and Claire Managan scored results; Dan Bushey helped with earlier versions of python scripts to collate sleep assay datasets. Jennifer Jeter imaged and scored EASI-FISH experiments. Janelia's FlyLight Project Team and Project Pipeline Support team, especially Allison Vannan, Jennifer Jeter, Joanna Hausenfluck, Zachary Dorman, Kelley Lee, and Geoffrey Meissner, performed CNS dissections, staining, and imaging. Janelia's Invertebrate Shared Resource and Scientific Computing contributed to stock generation and image processing, respectively. Geoffrey Meissner and Rob Svirskas contributed to the split-GAL4 website. Michael Kunst, Preeti Sareen, and Michael Nitabach contributed to early screening of split-GAL4 lines for effects on sleep. Wyatt Korff helped with establishment of the 96-well sleep assay. Heather Dionne, Martin Reyes, Anisha Ali, and Matthew Finger helped with conducting sleep experiments and analyzing data. We thank Brad Hulse, Alexander Bates, Gabi Maimon, Maria de la Paz, Dick Nässel, Mubarak Hussain Syed, Meet Zandawala, and members of the Rosbash lab for comments on earlier drafts of the manuscript and for helpful discussion. This work was supported by the Howard Hughes Medical Institute, NIH 2R15GM125073-03 (to DS), and NSF IOS 2042873 (to DS).

## Additional information

### Funding

| Funder | Grant reference number | Author |
|---|---|---|
| Howard Hughes Medical Institute | | Tanya Wolff<br>Mark Eddison<br>Nan Chen<br>Aljoscha Nern<br>Gerald M Rubin |

| Funder | Grant reference number | Author |
|---|---|---|
| National Institutes of Health | NIH 2R15GM125073-03 | Preeti Sundaramurthi Divya Sitaraman |
| National Science Foundation | IOS 2042873 | Preeti Sundaramurthi Divya Sitaraman |

The funders had no role in study design, data collection and interpretation, or the decision to submit the work for publication.

## Author contributions

Tanya Wolff, Conceptualization, Data curation, Formal analysis, Validation, Investigation, Methodology, Writing – review and editing, Conceived the study, created split-GAL4 lines and assigned them to cell CX types, and scored EASI-FISH experiments; Mark Eddison, Conceptualization, Data curation, Formal analysis, Validation, Investigation, Methodology, Writing – review and editing, Conceived the study and designed, conducted, and scored EASI-FISH experiments; Nan Chen, Data curation, Formal analysis, Validation, Investigation, Methodology, Designed and conducted behavioral assays; Aljoscha Nern, Validation, Investigation, Methodology, Writing – review and editing, Created split-GAL4 lines; Preeti Sundaramurthi, Investigation, Conducted behavioral assays; Divya Sitaraman, Conceptualization, Data curation, Formal analysis, Supervision, Funding acquisition, Validation, Investigation, Methodology, Writing – original draft, Writing – review and editing, Conceived the study and designed, conducted and interpreted behavioral assays; Gerald M Rubin, Conceptualization, Data curation, Formal analysis, Supervision, Funding acquisition, Validation, Investigation, Writing – original draft, Project administration, Conceived the study, created split-GAL4 lines, and wrote the paper with input from all authors

## Author ORCIDs

Divya Sitaraman ⓘ https://orcid.org/0000-0003-1197-0355
Gerald M Rubin ⓘ https://orcid.org/0000-0001-8762-8703

Reviewer #1 (Public review): https://doi.org/10.7554/eLife.104764.3.sa1
Reviewer #2 (Public review): https://doi.org/10.7554/eLife.104764.3.sa2
Author response https://doi.org/10.7554/eLife.104764.3.sa3

# Additional files

## Supplementary files

MDAR checklist

Supplementary file 1. Table of additional split-GAL4 lines organized by CX structure and cell type.

Supplementary file 2. Thermogenetic screen of selected stable-split lines in males.

Supplementary file 3. Thermogenetic screen of selected stable-split lines in females.

Supplementary file 4. Optogenetic screen of selected stable-split lines in males.

Supplementary file 5. Optogenetic screen of selected stable-split lines in females.

## Data availability

Original confocal stacks for imaging data can be downloaded from https://www.janelia.org/split-gal4 or https://flylight-raw.janelia.org/. Fly stocks are either available from the Bloomington Drosophila Stock Center or the authors as indicated at https://www.janelia.org/split-gal4. RNA-seq data are available at GEO under accession code GSE271123.

The following dataset was generated:

| Author(s) | Year | Dataset title | Dataset URL | Database and Identifier |
|---|---|---|---|---|
| Wolff T, Eddison M, Chen N, Nern A, Sundaramurthi P, Sitaraman D, Rubin GM | 2024 | Cell-type-specific genetic tools for the *Drosophila* central complex and their use to investigate neuropeptide expression and sleep regulation | https://www.ncbi.nlm.nih.gov/geo/query/acc.cgi?acc=GSE271123 | NCBI Gene Expression Omnibus, GSE271123 |

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
