## [Editor Report · eLife Assessment]

This is a **fundamental** body of work reporting anatomical, molecular, and functional mapping of the central complex in *Drosophila*. There were a few concerns of a minor nature, and all were addressed by the authors. The tools generated and the findings, which include characterization of neuromodulators used by different cells, will undoubtedly serve as a foundation for future studies of this brain region. The data are **compelling** and likely to have a major impact.

---

## [Referee Report · Reviewer #1 (Public review)]

Summary:

This work is meant to help create a foundation for future studies of the Central Complex, which is a critical integrative center in the fly brain. The authors present a systematic description of cellular elements, cell type classifications, behavioral evaluations and genetic resources available to the *Drosophila* neuroscience community.

Strengths:

The work contributes new, useful and systematic technical information in compelling fashion to support future studies of the fly brain. It also continues to set a high and transparent standard by which large-scale resources can be defined and shared.

Weaknesses:

Manuscript revisions by the authors addressed all proposed weaknesses from the original version.

---

## [Referee Report · Reviewer #2 (Public review)]

Summary:

In this paper, Wolff et al. describe an impressive collection of newly created split-GAL4 lines targeting specific cell types within the central complex (CX) of *Drosophila*. The CX is an important area in the brain that has been involved in the regulation of many behaviors including navigation and sleep/wake. The authors advocate that to fully understand how the CX functions, cell-specific driver lines need to be created. In that respect, this manuscript will be of very important value to all neuroscientists trying to elucidate complex behaviors using the fly model. In addition, and providing a further very important finding, the authors went on to assess neurotransmitter/neuropeptides and their receptors expression in different cells of the CX. These findings will also be of great interest to many and will help further studies aimed at understanding the CX circuitries. The authors then investigated how different CX cell types influence sleep and wake. While the description of the new lines and their neurochemical identity is excellent, the behavioral screen seems to be unfinished and could have been more matured.

Strengths:

(1) The description of dozens of cell-specific split-GAL4 lines is extremely valuable to the fly community. The strength of the fly system relies on the ability to manipulate specific neurons to investigate their involvement in a specific behavior. Recently, the need to use extremely specific tools has been highlighted by the identification of sleep-promoting neurons located in the VNC of the fly as part of the expression pattern of the most widely used dorsal-Fan Shaped Body (dFB) GAL4 driver. These findings should serve as a warning to every neurobiologist, make sure that your tool is clean. In that respect, the novel lines described in this manuscript are fantastic tools that will help the fly community.

(2) The description of neurotransmitter/neuropeptides expression pattern in the CX is of remarkable importance and will help design experiments aimed at understanding how the CX functions.

Weaknesses:

(1) I find the behavioral (sleep) screen of this manuscript to be incomplete. It appears to me that this part of the paper is not as developed as it could be. The authors have performed neuronal activation using thermogenetic and/or optogenetic approaches. For some cell types, only thermogenetic activation is shown. There is no silencing data and/or assessment of sleep homeostasis or arousal threshold. The authors find that many CX cell types modulate sleep and wake but it's difficult to understand how these findings fit one with the other. It seems that each CX cell type is worthy of its own independent study and paper. I am fully aware that a thorough investigation of every CX neuronal type in sleep and wake regulation is a herculean task. So, altogether I think that this manuscript will pave the way for further studies on the role of CX neurons in sleep regulation.

(2) Linked to point 1, it is possible that the activation protocols used in this study are insufficient for some neuronal types. The authors have used 29{degree sign} for thermogenetic activation (instead of the most widely used 31{degree sign}) and a 2Hz optogenetic activation protocol. The authors should comment on the fact that they may have missed some phenotypes by using these mild activation protocols.

(3) There are multiple spelling errors in the manuscript that need to be addressed.

Comments on revisions:

I am satisfied with the authors response. This paper provides excellent starting points for additional studies into the role of different CX cell types in sleep and wake.

---

## [Author Response]

The following is the authors’ response to the original reviews.

**Public Reviews:**

**Reviewer #1 (Public review):**
Summary:This work is meant to help create a foundation for future studies of the Central Complex, which is a critical integrative center in the fly brain. The authors present a systematic description of cellular elements, cell type classifications, behavioral evaluations and genetic resources available to the *Drosophila* neuroscience community.Strengths:The work contributes new, useful and systematic technical information in compelling fashion to support future studies of the fly brain. It also continues to set a high and transparent standard by which large-scale resources can be defined and shared.Weaknesses:manuscript p. 1"The central complex (CX) of the adult *Drosophila melanogaster* brain consists of approximately 2,800 cells that have been divided into 257 cell types based on morphology and connectivity (Scheer et al., 2020; Hulse et al. 2021; Wolff et al., 2015)."The 257 accumulated cell types have informational names (e.g., PBG2‐9.s‐FBl2.b‐NO3A.b) in addition to their associations with specific Gal4 lines and specific EM Body IDs. All this is very useful. I have one suggestion to help a reader trying to get a "bird's eye view" of such a large amount of detailed and multi-layered information. Give each of the 257 CX cell types an arbitrary number: 1 to 257. In fact, Supplemental File 2 lists ~277 cell types each with a number in sequence, so perhaps in principle, it is there. This could expedite the search function when a reader is trying to cross-reference CX cell type information from the text, to the Figures and/or to the Supplemental Figures. Also, the use of (arbitrary) cell type numbers could expedite the explanation of which cell types are included in any compilation of information (e.g., which ones were tested for specific NT expression).

In this report we adhered to the nomenclature introduced in Hulse et al. 2021. We agree that the nomenclature of cell types in the CX is imperfect. There are inherent limitations to what can be done with present data. Even between the hemibrain and FAFB/Flywire EM datasets, it was not possible to derive a one-to-one correspondence in many cases, largely because we do not yet have enough information to distinguish between natural variation within a cell type and distinct cell types (see Schlegel et al. 2024). Moreover, many cell type distinctions depend on connectivity differences that are observable only in EM datasets but not in LM images. Several research groups are currently engaged in a comprehensive and collaborative effort to update the CX nomenclature that will extend over the next few months as additional connectomes become available. This work will require hundreds of hours of effort from anatomical and computational experts in multiple laboratories who have a strong interest in the CX. Since the correspondence between the established Hulse et al nomenclature we use and this new nomenclature will be made clear, it will be easy to transfer our data to that new nomenclature. For all these reasons, we believe we should not unilaterally introduce any new naming systems at this time.

manuscript p 2"Figure 2 and Figure 2-figure supplements 1-4 show the expression of 52 new split-GAL4 lines with strong GAL4 expression that is largely limited to the cell type of interest. .... We also generated lines of lesser quality for other cell types that in total bring overall coverage to more than three quarters of CX cell types."This section describes the generation and identification of specific split Gal4 lines, and the presentation is generally excellent. It represents an outstanding compendium of information. My reading of the text suggests ~200 cell types have Gal4 lines that are of immediate use (having high specificity or v close-to-high). Use of an arbitrary number system (mentioned above) could augment that description for the reasons stated. For example, which of the 257 cell types are represented by split Gal4 lines that constitute the ~1/3 representing "high-quality lines "? A second comment relates to this study 's functional analysis of the contributions of CX cell types to sleep physiology. The recent literature contains renewed interest in the specific expression patterns of Gal4 lines that can promote sleep-like behaviors. In particular Gal4 line expression outside the brain (in the VNC and outside the CNS) have been raised as important elements that need be included for interpretation interpretation of sleep regulation. This present study offers useful information about a large number of expression patterns, as well as a basis with which to seek additional information., including mention of VNC expression in many cases However, perhaps I missed it, but I could not find a short description of the over-all strategy used to describe the expression patterns and feel that could be helpful. Were all Gal4 lines studied for expression in the VNC? and in the peripheral NS? It is probably published elsewhere, but even a short reprise would still be useful.

We added a couple of sentences to clarify that the lines were imaged in the adult female brain and VNC and many were also imaged in males. These data, including the ability to download the original confocal stacks, are contained in an on-line web source cited in the text. We also make clear that we did not assay expression outside of the brain, optic lobes and VNC. Therefore, we cannot rule out expression in the peripheral nervous system (other than detected in the axons of sensory neurons in the CNS) or in muscle or other non-neuronal cell types.

manuscript p 9Neurotransmitter expression in CX cell types"To determine what neurotransmitters are used by the CX cell types, we carried out fluorescent in situ hybridization using EASI-FISH (Eddison and Irkhe, 2022; Close et al., 2024) on brains that also expressed GFP driven from a cell-type-specific split GAL4 line. In this way, we could determine what neurotransmitters were expressed in over 100 different CX cell types based on ...."Reading this description, I was uncertain whether the >100 cell types mentioned were tested with all the NT markers by EASI-FISH? Also, assigning arbitrary numbers to the cell types (same suggestion as above) could help the reader more readily ascertain which were the ~100 cell types classified in this context.

The specific probes used for each cell type are indicated in Figure 9 and in Supplemental File 1.

manuscript p 10"Our full results are summarized below, together with our analysis of neuropeptide expression in the same cell types."I recommend specifying which Figures and Tables contain the "full results" indicated.

We changed the wording to read:

“Our full results are summarized, together with our analysis of neuropeptide expression in the same cell types, in Figures 5 -9 and in Supplemental File 1.”

NP expression in CX cell typesSimilar to the comments regarding studies of NT expression: were all ~100 cell types tested with each of the 17 selected NPs? Arbitrary numerical identifies could be useful for the reader to determine which cell types/ lines were tested and which were not yet tested.

We expanded the description in Methods to now read:

“For neurotransmitters, the specific probes used for each cell type are indicated in Figure 9 and in Supplemental File 1. For neuropeptides, each of the 17 selected NP probes shown in Figure 5—figure supplement 1 was used on all cell types in Figure 9 except those marked by “—” in the neuropeptide column.”

manuscript p. 11"The neuropeptide expression patterns we observed fell into two broad categories."This section presents information that is extensive and extremely useful. It supports consideration of peptidergic cell signaling at a circuits level and in a systematic fashion that will promote future progress in this field. I have two comments. First, regarding the categorization of two NP expression patterns, discernible by differences in cell number: this idea mirrors one present in prior literature. Recently the classification of the transcription factor DIMM summarizes this same two-way categorization (e.g., doi: 10.1371/journal.pone.0001896). That included the fact that a single NP can be utilized by cell of either category.

We inserted a sentence to acknowledge this earlier work:

“Such large neurosecretory cells often express the transcription factor DIMM (Park et al. 2008).”

Second, regarding this comment:"In contrast, neuropeptides like those shown in Figure 6 appear to be expressed in dozens to hundreds of cells and appear poised to function by local volume transmission in multiple distinct circuits."Signaling by NPs in this second category (many small cells) suggests more local diffusion, a smaller geographic expanse compared to "volume" signaling by the sparser larger peptidergic cells. Given this, I suggest re-consideration in using the term "volume" in this instance, perhaps in favor of "local" or "paracrine". This is only a suggestion and in fact rests almost entirely on speculation/ interpretation, as the field lacks a strong empirical basis to say how far NPs diffuse and act. A recent study in the fly brain of peptide co-transmitters (doi: 10.1016/j.cub.2020.04.025) provides an instructive example in which differences between the spatial extents of long-range (peptide 1) versus short-range (peptide 2) NP signaling may be inferred in vivo.

We have modified the text to now read:

“those shown in Figure 6 are expressed in dozens to hundreds of cells and appear poised to function by transmission to nearby cells in multiple distinct circuits.”

Spab was mentioned (Figure 6 legend) but discarded as a candidate NP to include based on a personal communication, as was Nplp1. The manuscript did not include reasons to do so, nor include a reference to spab peptide. I suggest including explicit reasons to discard candidate NPs.

While there is strong supportive evidence for many NPs in *Drosophila*, the fact that other transcripts express NPs is more circumstantial often relying simply on sequence analysis and without convincing evidence for a specific cognate receptor. We note that Spab is not listed as a neuropeptide in the current release of FlyBase. In these cases, we relied on the opinion of individuals with extensive experience in studying Drosophila NPs. The results obtained with the probes for Spab and Nplp1 are still available in Supplemental File 1.

In Fig 9-supplement 1, neurotransmitter biosynthetic enzymes were measured by RNA-seq for given CX cell types to augment the cell type classification. The same methods could be used to support cell type classification regarding putative peptidergic character (in Figure 9 supplement 2) by measuring expression levels of critical, canonical neuropeptide biosynthetic enzymes. These include the proprotein convertase dPC2 (amon); the carboxypeptidase dCPD/E (silver); and the amidating enzymes dPHM; dPal1; dPal2. PHM is most related to DBM (dopamine beta monooxygenase), the rate limiting enzyme for DA production, and greater than 90% of *Drosophila* neuropeptides are amidated. If the authors are correct in surmising widespread use of NPs by CX cell types (and I expect they are), there could be diagnostic value to report expression levels of this enzyme set across many/most CX cell types.

In our admittedly limited experience, most cells express these enzymes and the level we observed in confirmed NP expressing cell types was not reproducibly higher. (The complete data for all genes for the cell types we assayed are available from our deposition in the NCBI Gene Expression Omnibus with accession number GSE271123.) Given our small sample size we chose not to comment on this in the paper.

Comment #6Screen of effects on Sleep behaviorThis work is large in scope and as suggested likely presents excellent starting points for many follow-up studies. I again suggest assigning stable number identities to the elements described. In this case, not cell types, but split Gal4 lines. This would expedite the cross-referencing of results across the four Supplemental Files 3-6. For example, line SS00273 is entry line #27 in S Files 3 and 4, but line entry #18 in S Files 5 and 6.

We believe the interested reader can make this correspondence by searching the supplemental files which are excel spreadsheets. We note that both driver lines and cell types have stable identifiers that are used across Figures and Tables: the line numbers (for example, SS00273) for driver lines and the Hulse et al cell type names for cell types.

manuscript p 26Clock to CX"Not surprisingly, the connectome reveals that many of the intrinsic CX cell types with sleep phenotypes are connected by wired pathways (Figure 12 and Figure 12-figure supplement 1)."Do intrinsic CX cells with sleep phenotypes also connect by wired pathways to CX cells that do not have sleep phenotypes?

Yes, but we do not have high confidence that negative sleep phenotypes in our assays indicate no role in sleep.

"The connectome also suggested pathways from the circadian clock to the CX. Links between clock output DN1 neurons to the ExR1 have been described in Lamaze et al. (2018) and Guo et al. (2018), and Liang et al. (2019) described a connection from the clock to ExR2 (PPM3) dopaminergic neurons."The introduction to this section indicates a focus on connectome-defined synaptic contacts. Whereas the first two studies cited featured both physiological and anatomic evidence to support connectivity from clock cells to CX, the third did not describe any anatomical connections, and that connection may in fact be due to diffuse not synaptic signalingI could not easily discern the difference between Figs 12 and 12-S1? These appear to be highly-related circuit models, wherein the second features more elements. Perhaps spell out the basis for the differences between the two models to avoid ambiguity.

We clarify the supplemental diagram differs from the one in the main text by the inclusion of additional connections:

“The strongest of these connections are diagrammed in Figure 12, with Figure 12—figure supplement 1 also showing additional weaker connections.”

"...the cellular targets of Dh31 released from ER5 are unknown, however previous work (Goda et al., 2017; Mertens et al., 2005; Shafer et al., 2008) has shown that Dh31 can activate the PDF receptor raising the possibility of autocrine signaling."

Regarding pharmacological evidence for Dh31 activation of Pdfr: strong in vivo evidence was developed in doi: 10.1016/j.neuron.2008.02.018: a strong pdfr mutation greatly reduces response to synthetic dh31 in neurons that normally express Pdfr

We added the Shafer et al., 2008 reference.

manuscript p 30"Unexpectedly, we found that all neuropeptide-expressing cell types also expressed a small neurotransmitter."Did this conclusion apply only to CX cell types? - or was it also true for large peptidergic neurons? Prior evidence suggests the latter may not express small transmitters (doi: 10.1016/j.cub.2009.11.065). The question pertains to the broader biology of peptidergic neurons, and is therefore outside the strict scope of the main focus area - the CX. However, the text did initially consider peptidergic neurons outside the CX, so the information may be pertinent to many readers.

We did not look at other cell types in the current study and so cannot provide an answer.

**Reviewer #2 (Public review):**
Summary:In this paper, Wolff et al. describe an impressive collection of newly created split-GAL4 lines targeting specific cell types within the central complex (CX) of *Drosophila*. The CX is an important area in the brain that has been involved in the regulation of many behaviors including navigation and sleep/wake. The authors advocate that to fully understand how the CX functions, cell-specific driver lines need to be created. In that respect, this manuscript will be of very important value to all neuroscientists trying to elucidate complex behaviors using the fly model. In addition, and providing a further very important finding, the authors went on to assess neurotransmitter/neuropeptides and their receptors expression in different cells of the CX. These findings will also be of great interest to many and will help further studies aimed at understanding the CX circuitries. The authors then investigated how different CX cell types influence sleep and wake. While the description of the new lines and their neurochemical identity is excellent, the behavioral screen seems to be limited.Strengths:(1) The description of dozens of cell-specific split-GAL4 lines is extremely valuable to the fly community. The strength of the fly system relies on the ability to manipulate specific neurons to investigate their involvement in a specific behavior. Recently, the need to use extremely specific tools has been highlighted by the identification of sleep-promoting neurons located in the VNC of the fly as part of the expression pattern of the most widely used dorsal-Fan Shaped Body (dFB) GAL4 driver. These findings should serve as a warning to every neurobiologist, make sure that your tool is clean. In that respect, the novel lines described in this manuscript are fantastic tools that will help the fly community.(2) The description of neurotransmitter/neuropeptides expression pattern in the CX is of remarkable importance and will help design experiments aimed at understanding how the CX functions.Weaknesses:(1) I find the behavioral (sleep) screen of this manuscript to be limited. It appears to me that this part of the paper is not as developed as it could be. The authors have performed neuronal activation using thermogenetic and/or optogenetic approaches. For some cell types, only thermogenetic activation is shown. There is no silencing data and/or assessment of sleep homeostasis or arousal threshold. The authors find that many CX cell types modulate sleep and wake but it's difficult to understand how these findings fit one with the other. It seems that each CX cell type is worthy of its own independent study and paper. I am fully aware that a thorough investigation of every CX neuronal type in sleep and wake regulation is a herculean task. So, altogether I think that this manuscript will pave the way for further studies on the role of CX neurons in sleep regulation.(2) Linked to point 1, it is possible that the activation protocols used in this study are insufficient for some neuronal types. The authors have used 29{degree sign} for thermogenetic activation (instead of the most widely used 31{degree sign}) and a 2Hz optogenetic activation protocol. The authors should comment on the fact that they may have missed some phenotypes by using these mild activation protocols.

Our primary goal was to test the feasibility of using these tools in assessing sleep and wake function of neurons within the CX. In the process we uncovered several new neurons within the DFB-EB network that control sleep and make connections with previously identified sleep regulating neurons. For all single cell type lines and lines with sparse patterns and no VNC expression we present both optogenetics and thermogenetic data. The lines for which we only have thermogenetic but no optogenetic data are those which have multiple cell types or VNC expression. We felt that optogenetic data for these non-specific or contaminated lines would not reliably indicate a role for individual cell types in sleep regulation.

Many previous studies that have used 31 degrees have done so for shorter durations and often using different times of the day for manipulations. The lack of consistency between studies using this temperature may be due in part to the fact that 31 degrees alters behaviors of flies (including controls) and, for this reason, is usually not used for 24-hour activation durations.

To keep the screen consistent and ensure we capture changes in both daytime and nighttime sleep we used 29 degrees. The behavior of control flies is not as disrupted or altered at this temperature, and 29 degrees for activation is routinely used in behavioral experiments.

We similarly selected an optogenetic stimulation protocol that minimizes the response of flies to the red-light pulses. We chose this protocol because we found, in earlier experiments in a different project, that this level of stimulation was able to elicit activation phenotypes across a range of cell types (including several known clock neurons). However, we cannot rule out false negatives in both the TrpA and optogenetic experiments and agree that we might have missed some phenotypes.

Finally, as the reviewer rightfully points out, a thorough, detailed investigation of each cell type is a herculean task. We screened in both genders with very sparse, and often cell-type-specific, driver lines while using two distinct modes of activation and different methods for assessing sleep. For these reasons, we believe the GAL4 lines we identified provide excellent starting points for the additional investigations that will be required to better understand the roles of specific cell types.

(3) There are multiple spelling errors in the manuscript that need to be addressed.
**Reviewer #3 (Public review):**
Summary:The authors created and characterized genetic tools that allow for precise manipulation of individual or small subsets of central complex (CX) cell types in the *Drosophila* brain. They developed split-GAL4 driver lines and integrated this with a detailed survey of neurotransmitter and neuropeptide expression and receptor localization in the central brain. The manuscript also explores the functional relevance of CX cell types by evaluating their roles in sleep regulation and linking circadian clock signals to the CX. This work represents an ambitious and comprehensive effort to provide both molecular and functional insights into the CX, offering tools and data that will serve as a critical resource for researchers.Strengths:(1) The extensive collection of split-GAL4 lines targeting specific CX cell types fills a critical gap in the genetic toolkit for the *Drosophila* neuroscience community.(2) By combining anatomical, molecular, and functional analyses, the authors provide a holistic view of CX cell types that is both informative and immediately useful for researchers across diverse disciplines.(3) The identification of CX cell types involved in sleep regulation and their connection to circadian clock mechanisms highlights the functional importance of the CX and its integrative role in regulating behavior and physiological states.(4) The authors' decision to present this work as a single, comprehensive manuscript rather than fragmenting it into smaller publications each focusing on separate central complex components is commendable. This decision prioritizes accessibility and utility for the broader neuroscience community, which will enable researchers to approach CX-related questions with a ready-made toolkit.Weaknesses:While the manuscript is an outstanding resource, it leaves room for more detailed mechanistic exploration in some areas. Nonetheless, this does not diminish the immediate value of the tools and data provided.Appraisal:The authors have succeeded in achieving their aims of creating well-characterized genetic tools and providing a detailed survey of neurochemical and functional properties in the CX. The results strongly support their conclusions and open numerous avenues for future research. The work effectively bridges the gap between genetic manipulation, molecular characterization, and functional assessment, enabling a deeper understanding of the CX's diverse roles.Impact and UtilityThis manuscript will have a significant and lasting impact on the field, providing tools and data that facilitate new discoveries in the study of the CX, sleep regulation, circadian biology, and beyond. The genetic tools developed here are likely to become a standard resource for *Drosophila* researchers, and the comprehensive dataset on neurotransmitter and neuropeptide expression will inspire investigations into the interplay between neuromodulation and classical neurotransmission.Additional ContextThe breadth and depth of the resources presented in this manuscript justify its publication without further modification. By delivering an integrated dataset that spans anatomy, molecular properties, and functional relevance, the authors have created a resource that will serve the neuroscience community for years to come.
**Recommendations for the authors:**

**Reviewing Editor:**
The reviewers suggest that a nomenclature, perhaps a numbering system, be adopted for different cell types and Gal4 drivers in order to facilitate reading of the manuscript and cross-referencing.

We agree that a comprehensive reanalysis of the CX nomenclature is in order, but it is premature for us to attempt that as part of this study. This is best done after additional connectomes are generated to help resolve the degree of variation in morphology and connectivity between the same cell in multiple animals.

**Reviewer #3 (Recommendations for the authors):**
The authors have characterized a large number of split-GAL4 drivers targeting individual or small subsets of CX cell types. This manuscript delivers a detailed anatomical, molecular, and functional mapping of the CX.By integrating data on neurotransmitters, neuropeptides, and their receptors, the authors provide a holistic view of CX cell types that will undoubtedly serve as a foundation for future studies.The use of these genetic tools to identify CX cell types affecting sleep, as well as those linking the circadian clock to the CX, represents a significant advance. These findings hint at the diverse and integrative roles of the CX in regulating both behavior and physiological states.The authors' decision to present this work as a single, comprehensive manuscript rather than fragmenting it into smaller publications each focusing on separate central complex components is commendable. This decision prioritizes accessibility and utility for the broader neuroscience community, which will enable researchers to approach CX-related questions with a ready-made toolkit.While the manuscript leaves room for further exploration and mechanistic studies, the breadth and depth of the resources presented are more than sufficient to justify publication in their current form.The data on neuropeptide and receptor expression patterns, especially the observation that all examined CX cell types co-express a small neurotransmitter, opens intriguing new avenues of inquiry into the interplay between classical neurotransmission and neuromodulation in this region.This manuscript has provided a much-needed resource for the *Drosophila* neuroscience community and beyond. This work will facilitate important discoveries in CX function, sleep regulation, circadian biology, and more.